# Greater increases in intratumoral apparent diffusion coefficients after chemoradiotherapy predict better overall survival of patients with cervical cancer

Erikka Holopainen[1,2]*, Olli Lahtinen[1,2], Mervi Könönen[1,3], Maarit Anttila[4,5], Ritva Vanninen[1,2], Auni Lindgren[4,5]

1 Department of Radiology, Kuopio University Hospital, Kuopio, Finland, 2 Institute of Clinical Medicine, School of Medicine, Clinical Radiology, University of Eastern Finland, Kuopio, Finland, 3 Department of Clinical Neurophysiology, Kuopio University Hospital, Kuopio, Finland, 4 Department of Gynecology and Obstetrics, Kuopio University Hospital, Kuopio, Finland, 5 Institute of Clinical Medicine, School of Medicine, Obstetrics and Gynecology, University of Eastern Finland, Kuopio, Finland

* erikka.holopainen@pshyvinvointialue.fi

**Data Availability Statement:** All relevant data are within the manuscript and its Supporting Information files.

## Abstract

### Purpose

To evaluate whether 1) the intratumoral apparent diffusion coefficients (ADCs) change during cervical cancer treatment and 2) the pretreatment ADC values or their change after treatment predict the treatment outcome or overall survival of patients with cervical cancer.

### Methods

We retrospectively enrolled 52 patients with inoperable cervical cancer treated with chemoradiotherapy, who had undergone diffusion weighted MRI before treatment and post external beam radiotherapy (EBRT) and concurrent chemotherapy. A subgroup of patients (n = 28) underwent altogether six consecutive diffusion weighted MRIs; 1) pretreatment, 2) post-EBRT and concurrent chemotherapy; 3–5) during image-guided brachytherapy (IGBT) and 6) after completing the whole treatment course. To assess interobserver and intertechnique reproducibility two observers independently measured the ADCs by drawing freehand a large region of interest (L-ROI) covering the whole tumor and three small ROIs (S-ROIs) in areas with most restricted diffusion.

### Results

Reproducibility was equally good for L-ROIs and S-ROIs. The pretreatment ADCs were higher in L-ROIs (883 mm²/s) than in S-ROIs (687 mm²/s, $P < 0.001$). The ADCs increased significantly between the pretreatment and post-EBRT scans (L-ROI: $P < 0.001$; S-ROI: $P = 0.001$). The ADCs remained significantly higher than pretreatment values during the whole IGBT. Using S-ROIs, greater increases in ADCs between pretreatment and post-EBRT MRI predicted better overall survival ($P = 0.018$).

**Funding:** E.H: Radiological Society of Finland, 2021, https://sry.fi E.H: Abdominal Radiology Society of Finland, 2021, https://vatsaradiologit.sry.fi E.H: Kuopio University Research Foundation, 2021, https://www.psshp.fi/tutkimussaatio E.H: Grands from Kuopio University Hospital (Special government funding (VTR), grand 5063542) The funders had no role in study design, data collection and analysis, decision to publish or preparation of the manuscript.

**Competing interests:** The authors have declared that no competing interests exist.

## Conclusion

ADC values significantly increase during cervical cancer treatment. Greater increases in ADC values between pretreatment and post-EBRT predicted better overall survival using S-ROIs. Standardized methods for timing and delineation of ADC measurements are advocated in future studies.

## Introduction

Cervical cancer is the fourth most common cancer in women [1, 2]. Every year approximately 265,000 women die because of cervical cancer [3]. The prognosis, in terms of 5-year disease-free survival, ranges from nearly 100% for patients with Federation of Gynecology and Obstetrics (FIGO) stage IA cervical cancer to 5%–15% for stage IV [4]. Although the incidence and mortality of cervical cancer have decreased since the development of the Papanicolaou smear test in the 1950s [5, 6], it remains a major health problem that particularly affects women under the age of 40 years [7]. The majority (85%) of cervical cancers are squamous cell carcinomas (SCC), whereas other histological subtypes (adenocarcinoma, adenosquamous, undifferentiated) account for 15% of cases [5, 6].

Staging of cervical cancer has long relied on clinical findings alone, but since 2018, radiological imaging was added to the staging protocol [1, 2]. Magnetic resonance imaging (MRI) has been adopted in the main oncological imaging guidelines [8]. This is the most sensitive and specific imaging modality for initial staging and follow-up, and for assessing tumor responses and recurrence. Furthermore, MRI provides superior anatomical differentiation and soft tissue contrast resolution compared with computed tomography (CT) [1], is well correlated with assessments of parametrial invasion, and is useful in treatment planning [9]. Patients with early-stage disease, a tumor confined to the cervix, or tumors <4 cm in size (FIGO stages IB1, IB2 and IIA1) are usually treated with primary surgical resection and lymphadenectomy, but in cases with contraindications for surgery or anesthesia, chemoradiotherapy is an equally good alternative. Patients with more advanced disease (FIGO stage IB3 or greater) are treated with chemoradiotherapy [4, 5, 6, 10].

Diffusion-weighted imaging (DWI) is an essential part of the MRI protocol and can be used to improve tumor detection, staging and response to treatment [3, 8]. Diffusion reflects the effective displacement of water molecules freely moving through random Brownian motion over a given time [11, 12]. The restriction of the movement of water molecules depends on tissue cellularity, cell membrane integrity, and fluid viscosity. In tumors the increased tissue cellularity restricts the movement of water, which can be quantified by calculation of the apparent diffusion coefficient (ADC). In general, tumors have a greater DWI signal intensity and lower ADC than adjacent normal tissue [8, 9]. Low ADC values are usually found in highly dense tissues such as tumors, lymph nodes and fibrosis. In patients with various pelvic tumors (e.g., prostate, bladder, rectum, cervix) the ADC values could help us identify patients at risk of recurrence or with poorer prognosis and it can be used monitoring treatment response after chemoradiotherapy if the pre-treatment ADC values are compared to post-treatment values [13, 14]. However, there are no standardized methods for measuring ADCs in patients with cervical cancer. Consequently, various methods have been used to place regions of interest (ROI) [15, 16].

In this retrospective study, our aims were to investigate 1) whether different ROI sizes and placements affect the ADCs and 2) whether ADC values change during different phases of the

treatment protocol and, if so, 3) whether these changes are associated with the outcomes or survival of patients with cervical cancer.

## Materials and methods

### Study protocol and patients

We reviewed the records of consecutive patients with histologically proven cervical cancer diagnosed at our institution between 2009 and 2020. The study was approved by the Institutional Ethics Board of Kuopio University Hospital, which waived the need to obtain written consent from the patients due to the retrospective study design. Altogether 122 patients staged by pelvic MRI and body CT/PET-CT who were unsuitable for surgical treatment according to the FIGO 2018 staging system received concurrent chemoradiotherapy (CCRT) comprising external-beam radiotherapy (EBRT) (total dose: 45 Gy) with concurrent cisplatin-based chemotherapy and four cycles of image-guided brachytherapy (IGBT) according to the European Society of Gynecological Oncology (ESGO) guidelines [17] and were evaluated as candidates for the present study.

Altogether 52 patients (mean age 56 years) had undergone diffusion weighted MRI examinations before treatment and post- EBRT and concurrent chemotherapy and were included in the study.

A subgroup of those patients (n = 28, mean age 53 years) underwent consecutive diffusion weighted MRI examinations at six timepoints; 1) pretreatment, 2) post-EBRT and concurrent chemotherapy; 3–5) during IGBT, and 6) follow-up scan 3 months after the whole treatment (Fig 1).

### Imaging protocol and image analysis

All patients underwent pelvic MRI including unenhanced axial DWI, axial, coronal and sagittal T2WI and axial T1WI with a body array coil on a 1.5 T MRI scanner (Siemens MAGNETOM Avanto/Avanto Fit/Aera/Sola/Symphony Tim, Siemens Healthcare GmBH, Germany, or GE Signa Artist/Discovery MR450, GE Healthcare, USA). Due to the retrospective nature of the study, the MRI parameters were not standardized. Low $b$-value images of 50 s/mm$^2$ and high $b$-value axial images of 800 s/mm$^2$ were available for all patients. Additional $b$-value images of 0, 200, 400, 1000, 1200, or 1400 s/mm$^2$ were available for some patients. The DWI sequence protocols are described in S1 Table.

Two radiologists (EH and OL, with 3 and 6 years of experience of gynecological MRI, respectively) independently measured the ADCs. Observer 1 measured all DWI examinations. To evaluate reproducibility, observer 2 measured all 52 pretreatment MRIs and two thirds of post-EBRT, IGBT and follow-up MRIs in the subgroup. Sectra Workstation IDS7 PACS ROI tool (version 22.2.7, 2020, Sectra AB, Linköping, Sweden) was used to obtain the following automatically generated parameters for each ROI: mean; deviation; minimum and maximum ADCs; number of pixels; and area (mm$^2$). A large ROI (L-ROI) covering the whole tumor was manually drawn (freehand) on a single axial slice that showed the tumor at its maximum diameter and three circular ROIs with a diameter of 5 mm were placed on the regions of the tumor showing greatest restriction on the same slice visually assessed by both radiologists. S-ROIs could partly overlap in cases with very small lesions. Necrotic and cystic parts of the lesions were avoided. T2-weighted images and a localization tool were used for reference. Fig 2 illustrates the ADC measurement protocol on pretreatment images and Fig 3 shows the MRI findings at the six different time points.

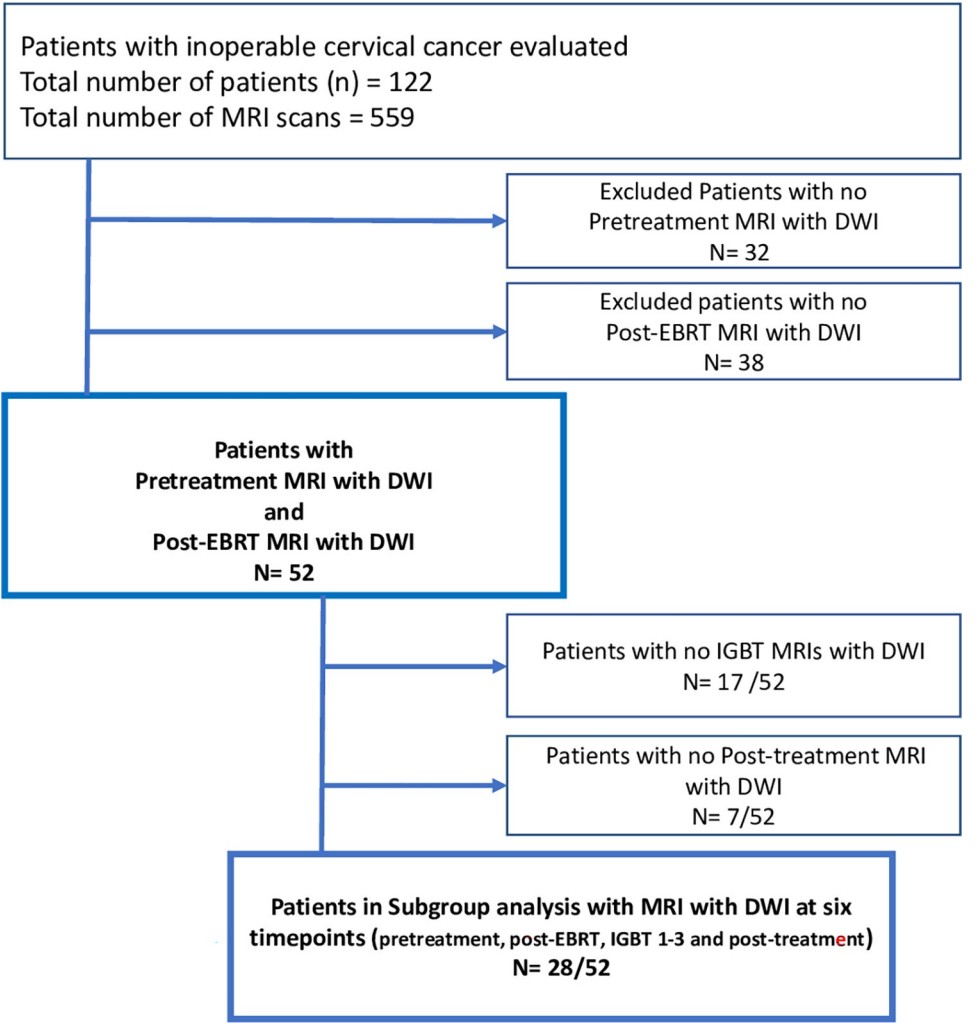

**Fig 1. Patient flowchart.** DWI = diffusion-weighted imaging; EBRT = external bean radiotherapy; IGBT = image-guided brachytherapy; MRI = magnetic resonance imaging.

## Statistical analysis

IBM SPSS Statistics for Windows 10 (version 26.0, 2019, IBM Corp, Armonk, NY, USA) was used for all statistical analyses. A *P* value of ≤0.05 was used to indicate statistical significance. The mean (L-ROI$_{mean}$) and minimum (L-ROI$_{min}$) ADCs were used for the analyses of L-ROI, whereas the lowest mean (S-ROI$_{mean}$) and lowest minimum (S-ROI$_{min}$) ADCs were used for the analyses of S-ROI. The interobserver reproducibility of the ADCs was analyzed by calculating the intraclass correlation coefficient, where values of 0.0–0.20 indicate poor, 0.21–0.40 indicate fair, 0.41–0.60 indicate moderate, 0.61–0.80 indicate good, and 0.81–1.00 indicate excellent correlation [15]. The Kolmogorov–Smirnov and Shapiro–Wilk normality tests were used to test the normality of the ADC measurements. The paired samples *t*-test was used to analyze the changes in ADC between time points. The Mann–Whitney U test was used to analyze the associations between pretreatment and post-EBRT ADCs and dichotomized clinical variables (tumor size, grade, stage, histology, lymph nodes, parametria invasion, adjuvant therapy, treatment response, and tumor recurrence). The Kaplan–Meier log rank method was used to determine overall survival and recurrence-free survival. ROC-curve analysis was used

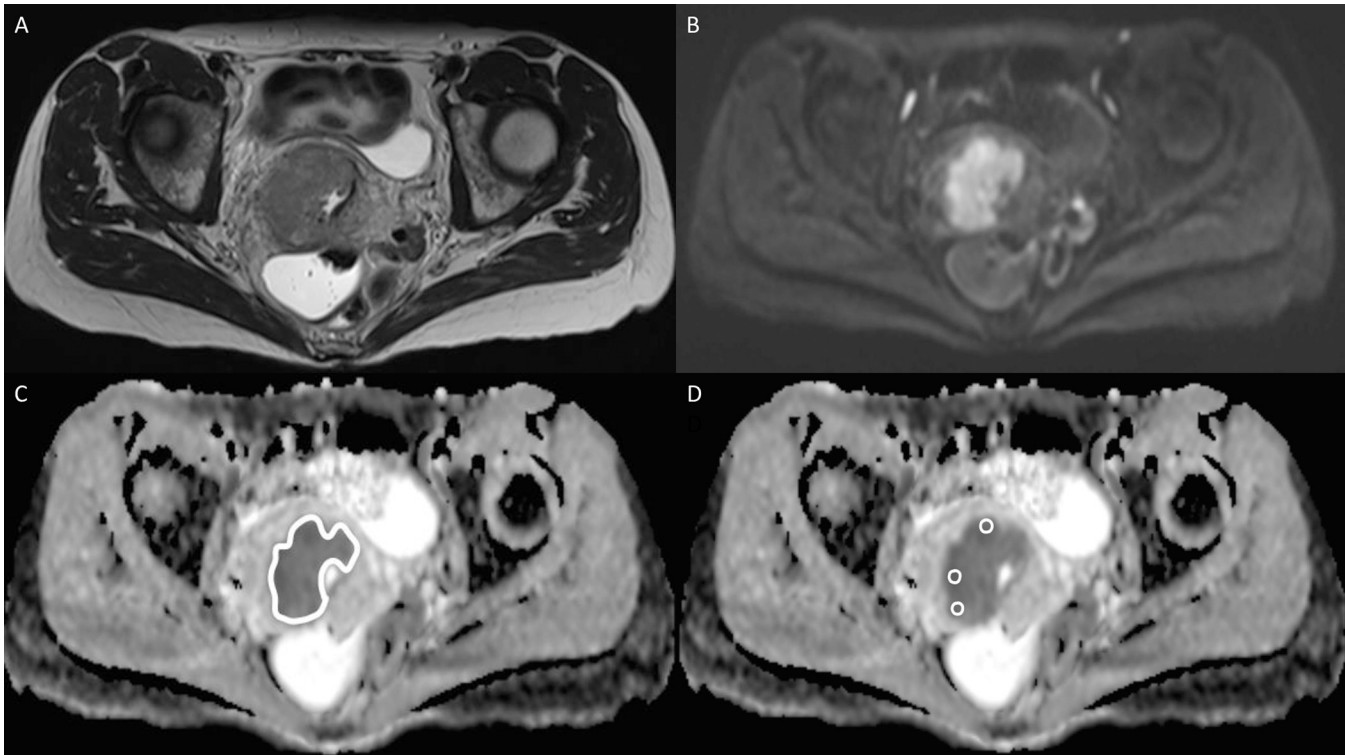

**Fig 2. MRI of a 55-year-old female with stage IIIc1 adenocarcinoma of the cervix.** (A) Axial T2-weighted image showing a solid mass, slightly hyperintense relative to the low signal of the cervical stroma. (B) The tumor appears bright on a high *b*-value diffusion-weighted image (*b*-value 800; B) and dark on the apparent diffusion coefficient (ADC) map (C). A large region of interest (L-ROI) was drawn freehand to cover the whole tumor on a single slice, with an ADC of 920 mm$^2$/s. (D) Three small ROIs (5 mm diameter) were drawn on the areas of the tumor showing the greatest restriction on the same slice, with an ADC of 807 mm$^2$/s. The patient had complete response after treatment.

to determine the Youden-index, which was used as a cut-off value for dichotomization. Cox regression analyses was used for multivariate analysis and to calculate the 1-year, 2-year, and 3-year overall and recurrence free survival.

## Results

### Patients

The final study population included 52 patients (mean age 55.7 years) of whom 44 were diagnosed with SCC, 6 with adenocarcinoma and 2 with carcinosarcoma. The mean diameter of the tumor at the time of diagnosis was 5.1 cm. The mean area of the L-ROI was 1225 mm$^2$ on pretreatment images, 379 mm$^2$ on post-EBRT images, and 277 mm$^2$ on post-treatment images. Twenty patients had cancer recurrence and 12 had progressive disease. The most common site for recurrence was lymph nodes following lungs and brain. During the follow up period from the time of diagnosis until end of year 2022, 17 patients had died. Twenty-eight patients underwent MRIs at all six timepoints and were included in the subgroup analysis. More detailed clinicopathological characteristics are shown in Table 1. Following EBRT and concurrent cisplatin chemotherapy, 37 (71%) of the 52 patients had a measurable residual tumor on the post-EBRT MRI while 15 (29%) of the tumors had vanished during the chemoradiotherapy. At the 3-month post-treatment scans, only 10 patients (19%) had a visible residual tumor on MRI.

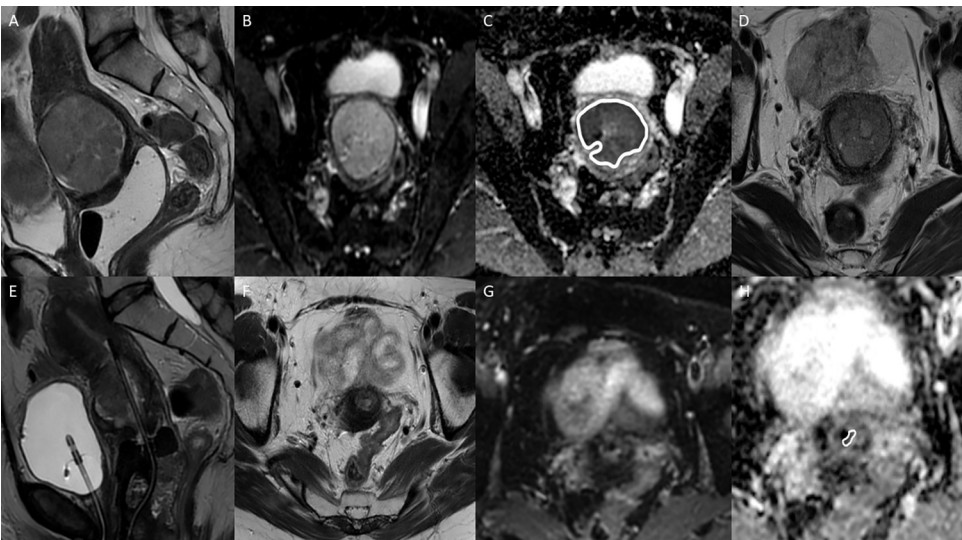

**Fig 3. MRI of a 55-year-old female with stage IIIc1 adenocarcinoma of the cervix.** (A) Pretreatment sagittal T2-weighted image showing a large solid tumor with a moderate T2-signal. The tumor appears bright on a high B-value diffusion weighted image (b 1000; B) and dark on the apparent diffusion coefficient (ADC) map (C), with an ADC for the large region of interest (L-ROI$_{mean}$) of 963 mm$^2$/s. (D) A large tumor is still visible after external beam radiotherapy. The ADC is 1057 mm$^2$/s, corresponding to an increase of 10%. (E) By the time of third imaging during image-guided brachytherapy, the tumor has reduced in size. (F, G, H) A small residual tumor is visible after treatment with an ADC of 625 mm$^2$/s. Despite the presence of a small residual tumor, the patient remained stable after treatment and no recurrence has been detected.

## ADCs and their changes during treatment

In the pretreatment MRI, the mean intratumoral ADC was 883 mm$^2$/s (range 401–1284) for L-ROI$_{mean}$ and was significantly lower 688 mm$^2$/s (range 372–1114, $P < 0.001$) for S-ROI$_{mean}$. The ADCs were significantly higher at the post-EBRT measurement both for L-ROI$_{mean}$ (1114 mm$^2$/s, $P < 0.001$) and S-ROI$_{mean}$ (970 mm$^2$/s, $P = 0.001$). Between pretreatment and post-EBRT MRI, the mean ADCs increased by 26% for L-ROI$_{mean}$ ($P < 0.001$), by 41% for S-ROI-$_{mean}$ ($P < 0.001$), by 55% for L-ROI$_{min}$ ($P < 0.001$) and by 47% for S-ROI$_{min}$ ($P < 0.001$). The ADCs remained significantly higher than pretreatment values during the whole IGBT (IGBT1: L-ROI$_{mean}$ 1244 mm$^2$/s, $P < 0.001$; S-ROI$_{mean}$ 1130 mm$^2$/s, $P < 0.001$; IGBT2: L-ROI$_{mean}$ 1058 mm$^2$/s, $P = 0.045$; S-ROI$_{mean}$ 960 mm$^2$/s, $P = 0.012$; IGBT3: L-ROI$_{mean}$ 1197 mm$^2$/s, $P = 0.001$; S-ROI$_{mean}$ 1082 mm$^2$/s, $P = 0.008$). To further analyze the importance of a residual tumor after treatments, the patients with a measurable tumor in the post-EBRT scans (n = 37) were divided to those with a residual tumor in the 3-month post-treatment scan (n = 10) and those with no residual tumor in the 3-month post-treatment scan (n = 27). The ADCs were significantly higher at the post-EBRT measurements than in the pretreatment scans in both groups (L-ROI$_{mean}$ $P = 0.006$, S-ROI$_{min}$ $P = 0.017$ in residual tumor group and L-ROI$_{mean}$ $P < 0.001$, S-ROI$_{min}$ $P < 0.001$ in no residual tumor group respectively. The ADC change was also significant for L-ROI$_{min}$ and S-ROI$_{mean}$ in both groups). In patients with residual tumor after IGBT, the ADCs at 3 moths' post-treatment MRI were significantly higher than pretreatment ADCs (L-ROI$_{min}$ 843 mm$^2$/s, $P = 0.002$, S-ROI$_{mean}$ 1001 mms$^2$/s, $P = 0.008$, S-ROI$_{min}$ 884 mm$^2$/s, $P = 0.006$) but there were no significant changes in the ADCs at 3 months post-treatment compared with post-EBRT. The mean ADC values and the changes in ADCs during and after treatment are shown in Figs 4 and 5.

**Table 1. Clinicopathological characteristics of the patients.** Characteristics of all patients with diffusion weighted MRIs before treatment and post-EBRT (n = 52) and the subgroup of them with MRIs at all six timepoints: pretreatment, post-EBRT and concurrent chemotherapy, during IGBT and after completing the whole treatment course (n = 28) are shown separately.

| Variable | All patients n = 52 | | Patients with 6 follow up MRIs n = 28 | |
|---|---|---|---|---|
| | n (%) | Mean | n (%) | Mean |
| Age (years) | | 55.7 ± 16.1 | | 52.5 ± 14.3 |
| Histopathology | | | | |
| Adenocarcinoma | 6 (11) | | 3 (20) | |
| SCC | 44 (85) | | 20 (80) | |
| Carcinosarcoma | 2 (4) | | | |
| Grade | | | | |
| 2 | 20 (39) | | 11 (48) | |
| 3 | 14 (27) | | 7 (30) | |
| Missing | 18 (34) | | 5 (22) | |
| Stage (FIGO 2018) | | | | |
| IB1 | 4 (8) | | 3 (13) | |
| IB2 | 3 (6) | | 2 (9) | |
| IB3 | 2 (4) | | | |
| IIB | 22 (42) | | 10 (44) | |
| IIIA | 1 (2) | | | |
| IIIB | 4 (8) | | 3 (13) | |
| IIIC1 | 4 (8) | | 2 (9) | |
| IIIC2 | 5 (10) | | 1 (4) | |
| IVA | 2 (4) | | 1 (4) | |
| IVB | 5 (10) | | 1 (4) | |
| Tumor size at diagnosis (cm) | | 5.1 ± 2.3 | | 5.0 ± 2.0 |
| Lymph nodes | | | | |
| Positive | 30 (58) | | 13 (57) | |
| Negative | 22 (42) | | 10 (44) | |
| Parametria invasion | | | | |
| Yes | 38 (73) | | 16 (70) | |
| No | 14 (27) | | 7 (30) | |
| Adjuvant therapy | | | | |
| Yes | 5 (10) | | 1 (4) | |
| No | 47 (90) | | 22 (96) | |
| Follow-up time (months) | | 29.9 ± 22.7 | | 41.0 ± 24.8 |
| Cancer recurrence | | | | |
| No | 32 (62) | | 15 (65) | |
| Yes | 20 (38) | | 8 (35) | |
| PD | 12 (23) | | 4 (17) | |
| Deceased | | | | |
| Yes | 17 (33) | | 6 (26) | |
| No | 35 (67) | | 17 (74) | |

SCC = squamous cell carcinoma, PD = progressive disease

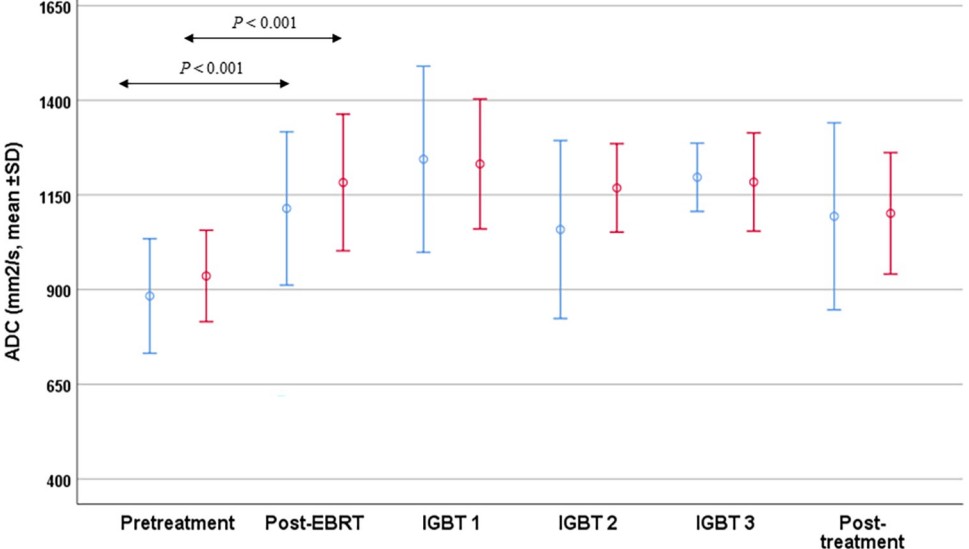

L-ROImean. Blue = observer 1, red = observer 2

**Fig 4. Mean apparent diffusion coefficients and their changes during treatment in whole tumor ROIs.** The measurements were derived from large regions of interest (L-ROIs) covering the whole tumor drawn by two independent observers. The graph shows the mean value for the L-ROI, with values for observer 1 in blue and those for observer 2 in red. ADC = apparent diffusion coefficient; EBRT = external beam radiotherapy; IGBT = image-guided brachytherapy; L-ROI$_{mean}$ = mean ADC for the large regions of interest; SD = standard deviation.

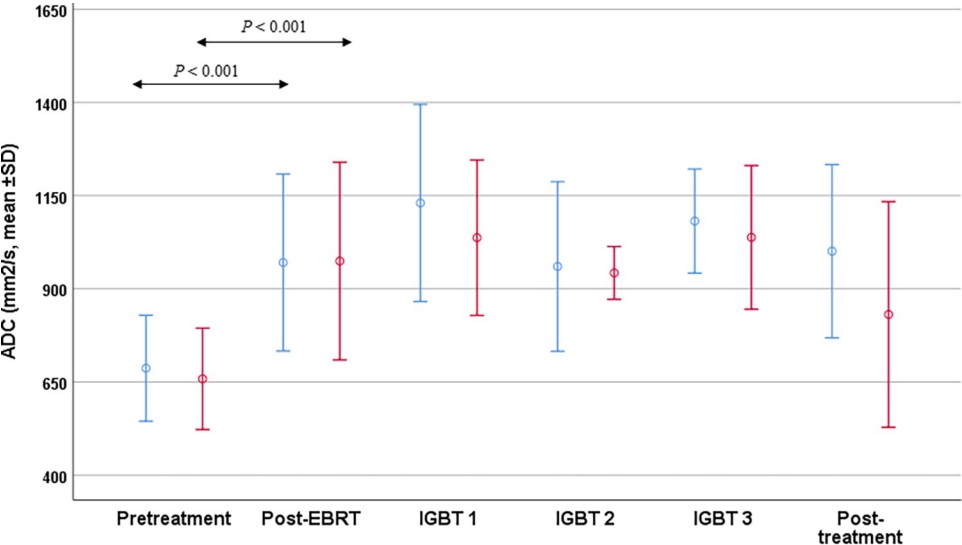

S-ROImean. Blue = observer 1, red = observer 2

**Fig 5. Mean apparent diffusion coefficients and their changes during treatment in small ROIs.** Measurements were taken from three small regions of interest (S-ROIs) with a diameter of 5 mm placed on areas of tumor showing greatest restriction by two independent observers. The graph shows the lowest mean value of the three regions. ADC = apparent diffusion coefficient; EBRT = external beam radiotherapy; IGBT = image-guided brachytherapy; S-ROI$_{mean}$ = mean ADC for the small regions of interest; SD = standard deviation.

**Table 2. Intraclass correlation coefficient of the ADC measurements between observers 1 and 2.**

|  | *n* of measurements for observers 1/2 | ICC | *P* |
|---|---|---|---|
| Pretreatment (n = 52) |  |  |  |
| L-ROI$_{mean}$ | 52/51 | 0.939 | <0.001 |
| L-ROI$_{min}$ | 52/51 | 0.947 | <0.001 |
| S-ROI$_{mean}$ | 52/51 | 0.914 | <0.001 |
| S-ROI$_{min}$ | 52/51 | 0.927 | <0.001 |
| Post-EBRT (n = 23) |  |  |  |
| L-ROI$_{mean}$ | 20/18 | 0.933 | <0.001 |
| L-ROI$_{min}$ | 20/18 | 0.925 | <0.001 |
| S-ROI$_{mean}$ | 20/18 | 0.667 | 0.015 |
| S-ROI$_{min}$ | 20/18 | 0.754 | 0.003 |
| IGBT1 |  |  |  |
| L-ROI$_{mean}$ | 14/14 | 0.774 | 0.008 |
| L-ROI$_{min}$ | 14/14 | 0.905 | <0.001 |
| S-ROI$_{mean}$ | 14/14 | 0.852 | 0.001 |
| S-ROI$_{min}$ | 14/14 | 0.876 | 0.001 |

L-ROI = large ROI; S-ROI = lowest ADC of three small ROIs; ROI = region of interest; mean = mean value; min = minimum value; EBRT = external beam radiotherapy; IGBT = image-guided brachytherapy; ICC = intraclass correlation coefficient, n = number of DWI scans evaluated by both observers.

### Reproducibility of the ADC measurements

The interobserver agreement between two observers was good to excellent for both ROI delineation methods. The intraclass correlation coefficients ranged from 0.77 to 0.93 for L-ROIs ($P < 0.001$) and from 0.67 to 0.92 for S-ROIs ($P < 0.001$ to 0.015) (Table 2).

### ADCs and prognostic factors for cervical cancer

Tumor grade or histology were not associated with the pretreatment ADCs. More advanced tumor stage tended to be associated ($P = 0.063$) with higher ADCs in L-ROI$_{mean}$. Larger tumors of over 4 cm had lower ADC values post EBRT in L-ROI$_{mean}$ ($P = 0.034$) and tended to be lower in S-ROI$_{mean}$ ($P = 0.076$). Tumors that recurred in follow up showed higher ADC values in pretreatment MRI (S-ROI$_{min}$, $P = 0.045$; L-ROI$_{min}$, $P = 0.055$). In patients, who had adjuvant therapy (n = 5) the increase in ADC values between pre- and post-EBRT did not reach statistical significance using two of the measurement methods (L-ROI$_{mean}$, $P = 0.084$; S-ROI$_{min}$, $P = 0.105$), while in patients with no adjuvant therapy (n = 47) the increase was highly significant with all ROI measurement methods ($P = 0.001$) (Table 3).

### Patient survival

The median follow-up time was 30 months (range 2–113 months). Thirty-two patients (62%) had no signs of cancer recurrence by the end of the follow-up period (from diagnosis until the end of 2022). Cancer recurrence was detected by imaging in 20 patients (38%) and 12 patients (23%) were diagnosed with progressive disease. The median recurrence-free survival was 3.5 months (range 0–18 months). At the end of the follow-up, 35 (67%) patients were alive and 17 (33%) were deceased. Five (9.6%) patients died within the first follow-up year, thirteen (25%) were deceased at 2-year timepoint and fourteen (27%) at 3-year timepoint.

An increase in the intratumoral ADC of >47% between the pre- and post-EBRT scans for S-ROI$_{min}$ predicted better overall survival ($P = 0.018$) in Kaplan Meier analyses (Fig 6).

**Table 3. Associations of intratumoral pretreatment and post-EBRT apparent diffusion coefficient (ADC) values with clinical variables and the significance of the changes in ADC values between time points.** All 52 patients underwent DWI post-EBRT, but only 37 patients had a measurable residual tumor.

| Variable | ADCs (mm²/s) | | | | | | | | Change after EBRT treatment *P†* (n = 37) | | | |
|---|---|---|---|---|---|---|---|---|---|---|---|---|
| Between-group *P*\* | Pretreatment (n = 52) | | | | Post-EBRT (n = 37/52 with residual tumor) | | | | | | | |
| | L-ROI | L-ROI | S-ROI | S-ROI | L-ROI | L-ROI | S-ROI | S-ROI | L-ROI | L-ROI | S-ROI | S-ROI |
| | mean | min | mean | min | mean | min | mean | min | mean | min | mean | min |
| All patients | 883.0 | 529.6 | 686.8 | 576.0 | 1114.1 | 822.1 | 970.4 | 847.3 | 0.001 | 0.001 | 0.001 | 0.001 |
| Grade | | | | | | | | | | | | |
| 2 | 853.5 | 544.9 | 695.0 | 598.9 | 1104.9 | 813.3 | 957.9 | 837.9 | 0.001 | 0.001 | 0.001 | 0.004 |
| 3 | 909.4 | 518.1 | 667.3 | 548.5 | 1116.2 | 881.7 | 974.4 | 858.3 | 0.031 | 0.010 | 0.020 | 0.032 |
| *P* | 0.564 | 0.462 | 0.433 | 0.195 | 0.617 | 0.958 | 0.781 | 0.781 | | | | |
| Stage | | | | | | | | | | | | |
| 1–2 | 855.8 | 529.6 | 677.3 | 564.2 | 1149.8 | 844.3 | 989.0 | 872.1 | 0.001 | 0.001 | 0.001 | 0.001 |
| 3–4 | 926.5 | 529.5 | 702.8 | 595.0 | 1055.4 | 785.6 | 937.5 | 803.3 | 0.002 | 0.001 | 0.001 | 0.005 |
| *P* | 0.063 | 0.985 | 0.682 | 0.829 | 0.121 | 0.364 | 0.420 | 0.449 | | | | |
| Histology | | | | | | | | | | | | |
| SCC | 886.7 | 521.5 | 686.8 | 502.0 | 1117.9 | 826.1 | 968.3 | 855.3 | 0.001 | 0.001 | 0.001 | 0.001 |
| Adenocarcinoma | 895.8 | 600.8 | 734.3 | 646.7 | 1131.2 | 864.0 | 1036.4 | 871.4 | 0.057 | 0.034 | 0.051 | 0.064 |
| *P* | 0.811 | 0.179 | 0.542 | 0.325 | 0.873 | 0.801 | 0.741 | 0.906 | | | | |
| Tumor size | | | | | | | | | | | | |
| <4 cm | 843.8 | 507.5 | 667.0 | 547.7 | 1257.0 | 912.1 | 1127.2 | 997.2 | 0.001 | 0.001 | 0.001 | 0.001 |
| ≥4 cm | 896.0 | 536.9 | 693.6 | 585.5 | 1068.1 | 793.1 | 918.0 | 797.3 | 0.001 | 0.001 | 0.001 | 0.001 |
| *P* | 0.363 | 0.759 | 0.443 | 0.466 | **0.034** | 0.620 | 0.076 | 0.165 | | | | |
| Response | | | | | | | | | | | | |
| CR | 877.8 | 519.9 | 679.5 | 556.5 | 1101.2 | 843.1 | 980.1 | 871.1 | 0.001 | 0.001 | 0.001 | 0.001 |
| PD | 900.3 | 561.8 | 710.3 | 641.2 | 1160.9 | 745.8 | 936.1 | 764.0 | 0.008 | 0.017 | 0.022 | 0.084 |
| *P* | 0.373 | 0.778 | 0.444 | 0.193 | 0.396 | 0.417 | 0.621 | 0.304 | | | | |
| Recurrence | | | | | | | | | | | | |
| No | 865.5 | 483.4 | 660.0 | 534.4 | 1115.6 | 844.2 | 989.9 | 881.2 | 0.001 | 0.001 | 0.001 | 0.001 |
| Yes | 911.0 | 603.4 | 728.3 | 642.5 | 1111.6 | 785.6 | 939.6 | 793.9 | 0.001 | 0.002 | 0.001 | 0.007 |
| *P* | 0.240 | 0.055 | 0.130 | **0.045** | 0.802 | 0.826 | 0.770 | 0.527 | | | | |
| Lymph nodes | | | | | | | | | | | | |
| Negative | 882.8 | 508.4 | 691.6 | 580.9 | 1165.6 | 835.5 | 1991.1 | 884.9 | 0.001 | 0.001 | 0.001 | 0.002 |
| Positive | 883.1 | 545.0 | 683.1 | 572.5 | 1078.9 | 812.9 | 948.4 | 820.4 | 0.001 | 0.001 | 0.001 | 0.001 |
| P | 0.684 | 0.493 | 0.690 | 0.919 | 0.228 | 0.951 | 0.665 | 0.574 | | | | |
| Parametria invasion | | | | | | | | | | | | |
| No | 866.7 | 572.4 | 712.3 | 611.8 | 1191.8 | 884.2 | 1069.4 | 967.0 | 0.001 | 0.001 | 0.001 | 0.001 |
| Yes | 889.0 | 513.8 | 677.1 | 562.8 | 1085.3 | 799.0 | 932.3 | 801.2 | 0.002 | 0.002 | 0.002 | 0.009 |
| P | 0.628 | 0.312 | 0.598 | 0.348 | 0.231 | 0.745 | 0.281 | 0.197 | | | | |
| Adjuvant therapy | | | | | | | | | | | | |
| Yes | 907.0 | 585.6 | 686.3 | 594.8 | 1081.2 | 722.4 | 904.0 | 742.0 | 0.084 | 0.040 | 0.006 | 0.105 |
| No | 880.4 | 523.6 | 686.8 | 574.0 | 1119.2 | 837.6 | 978.7 | 860.4 | 0.001 | 0.001 | 0.001 | 0.001 |
| P | 0.49.0 | 0.512 | 0.876 | 0.756 | 0.467 | 0.223 | 0.746 | 0.461 | | | | |

L-ROI = large ROI; S-ROI = lowest ADC of three small ROIs; ROI = region of interest; mean = mean value; min = minimum value; SCC = squamous cell cancer;

CR = complete response; PD = progressive disease

\*For the between-group differences in ADCs.

†For the within-group changes in ADCs between the two timepoints.

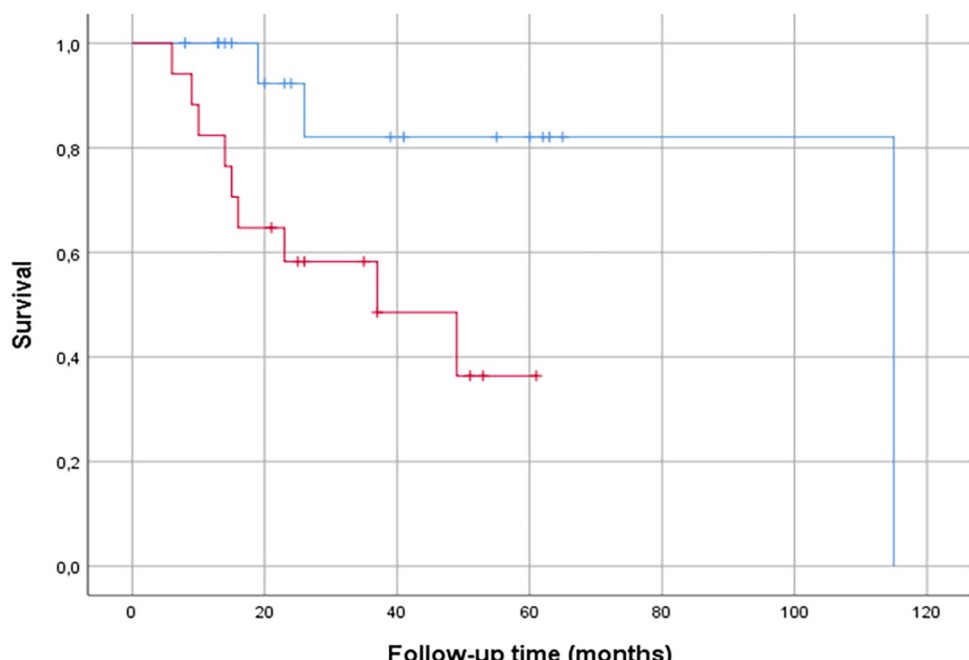

**Fig 6. Univariate analysis of cumulative overall survival according to the relative change in ADC values from pretreatment to post-EBRT MRI scan using intratumoral S-ROIs.** ADC value difference between the pretreatment and post-EBRT scans was dichotomized using ROC curve to define the Youden index providing the cutoff value of 197.5 mm$^2$/s. Blue line indicates group of patients with an increase of over 197.5 mm$^2$/s in ADC values (n = 14). Red line indicates group of patients with an increase of less than 197.5 mm$^2$/s in ADC values (n = 23) An increase in the ADC of over 197.5 mm$^2$/s for S-ROI$_{min}$ predicted better overall survival ($P$ = 0.018).

Because of the large variety of follow-up times, we calculated the survival analysis also using 1-year, 2-year, and 3-year timepoints. A higher pretreatment ADC value predicted better overall survival at 1-year timepoint (L-ROI$_{mean}$, $P$ = 0.05). A greater increase in the ADC values between pre-and post-EBRT scans predicted better overall survival at 2-year follow-up period (L-ROI$_{mean}$, $P$ = 0.045; S-ROI$_{min}$, $P$ = 0.008) and at 3-year follow-up period (S-ROI $_{min}$, $P$ = 0.024).

We included ADC values and their change, measured with different ROI delineation methods, together with known prognostic factors of cervical cancer (stage, tumor size, lymph node, parametria invasion, adjuvant therapy and residual tumor) to Cox regression multivariate analysis. For overall survival, at 1-year timepoint pretreatment ADC value tended to remain significant in multivariate analysis (L-ROI$_{mean}$ $P$ = 0.067, HR 37.515 (95%CI 0.78–1811.34). ADC value increase between pretreatment and post-EBRT measured with S-ROI$_{min}$ tended to be significant also in multivariate analysis for the whole follow-up time ($P$ = 0.063, HR 0.199 (95% CI 0.04–1.09) and for the 2-year timepoint ($P$ = 0.067, HR 0.137 (95% CI 0.02–1.15). None of the other prognostic factors remained statistically significant in multivariate analysis in our cohort.

In recurrence free survival analyses, a recurrence free survival was predicted by the greater ADC increase between pre- and post-EBRT using S-ROI$_{min}$ ($P$ = 0.042) at 1-year timepoint, using L-ROI$_{min}$ ($P$ = 0.034) and S-ROI$_{min}$ ($P$ = 0.033) at 2-year timepoint and using L-ROI$_{min}$ ($P$ = 0.012) and S-ROI$_{min}$ ($P$ = 0.043) at 3-year timepoint. Also, pretreatment ADC value S-ROI$_{min}$ was predictive for RFS at 3-year timepoint ($P$ = 0.049). Using Cox regression multivariate analysis ADC value change measured with LROI$_{min}$ was statistically significant at 3-year timepoint ($P$ = 0.033, HR 0.204 (95% CI 0.05–0.88) and pretreatment ADC value

S-ROI$_{min}$ at 3-year timepoint remained also significant ($P = 0.050$, HR 0.340 (95%CI 0.12–1.00). More detailed information about the overall and recurrence free survival analyses are shown in S2 and S3 Tables and S1 Fig.

## Discussion

We retrospectively evaluated 52 patients with inoperable cervical cancer who underwent concurrent chemoradiotherapy comprising external-beam radiotherapy and image-guided brachytherapy who had undergone diffusion weighted MRI as part of their diagnostic and follow-up clinical protocol. We examined whether the treatment induces increases in ADC values and whether these ADC values at different timepoints or their changes predict treatment outcomes and survival. We detected significant increases in the ADCs of the cervical tumor mass between the images obtained pretreatment, post-EBRT, and in subgroup analyses during IGBT. A greater increase in the ADC between the pre- and post-EBRT images predicted better overall survival. We also evaluated whether the ROI size and placement influenced the results. We chose to compare a single freehand L-ROI drawn on a slice with the maximum tumor diameter and three circular S-ROIs that were placed on the tumor regions with greatest restriction on the same slice.

Our results are parallel with those of several earlier studies [18–21]. Quantitative analysis of ADCs has been proven to be useful in diagnostic and prognostic assessments and for predicting the treatment outcomes and survival of patients with cervical cancer [22]. Naganawa et al. [20] and McVeigh et al. [21] reported that the mean ADC for cervical cancer is significantly lower than that of normal cervical tissue and that the mean ADC of tumors increases significantly after EBRT. In a meta-analysis of patients with cervical cancer, Fu et al. [23] reported that the mean ADCs were significantly greater after radiotherapy and chemotherapy than the pretreatment ADC values, and that ADCs may be effective for evaluating the treatment outcomes in patients with cervical cancer.

In our cohort, we observed significant increases in the ADC values from before treatment through to the end of treatment. Interestingly, the ADC values were somewhat lower after IGBT2 compared to those post-EBRT. Several of the tumors were so small in the IGBT scans that the ADC measurements could not be performed. In the subgroup analysis, only 14 tumors could be measured at IGBT compared to the 25 tumors measured post-EBRT, possibly leaving the most aggressive ones still visible. In addition, the tumors often shrink during IGBT, and the remaining smaller and denser tumors may in part explain the lower ADC values. Few studies have assessed the ADCs at six different time points. Thus, our study provides new information on the evolution of ADCs during a treatment protocol comprising EBRT and IGBT and might help to schedule the timing of follow-up MRI scans. Based on the results of this study, mid-treatment MRI should be performed before the residual tumor disappears for prognostic purposes, and based on the results of multivariate analysis, it should be performed immediately after EBRT. Prognosis can be predicted by calculating the ADC change value between pretreatment and post-EBRT.

Nakamura et al. [24] reported that the mean ADCs of the primary tumor were significantly associated with FIGO stage, tumor size, stromal and parametrial invasion, lymph node metastasis, lymph vascular space involvement, and adjuvant therapy. Lower ADC values were associated with higher grade, stage and parametrial invasion but they also tended to be associated with more complete treatment response [9]. In our cohort, tumor grade, size, histology, parametria invasion or lymph node metastasis were not associated with the pretreatment ADCs while a more advanced tumor stage tended to be associated with ADCs. The somewhat different results are possibly due to the inclusion of different patient populations, variations in ADC measurement techniques, and the small study population.

Erbay et al. reported that the pretreatment mean ADCs were significantly lower in patients with tumor recurrence [25] while in contrast Somoye et al. [26] reported that the pretreatment ADCs were not correlated with the prognosis of cervical cancer. Furthermore, Bae et al. [27] reported that pretreatment ADC was not associated with tumor recurrence but tumor ADC change between pretreatment and post-EBRT was a significant independent predictor of tumor recurrence after therapy. A recent systemic review and meta-analysis [28] revealed that the pretreatment ADCs cannot be used alone to predict the outcome of CCRT reliably in patients with cervical cancer. Meng et al. [29] and Bae et al. [27] found that the mid-treatment mean ADCs were significantly lower in patients with tumor recurrence. Our results suggest that the predictive information of ADC measurements is greater when the pretreatment and post-EBRT ADCs are compared, instead of using the ADC values at a single timepoint.

In our cohort, a recurrence free survival was predicted by the greater ADC increase between pre- and post-EBRT. Furthermore, in the Kaplan–Meier univariate analysis of cumulative overall survival, a greater increase (>47%) in the ADC between pre- and post-EBRT was associated with better overall survival ($P = 0.018$) when using S-ROI$_{min}$. A larger ADC change rate between pre-treatment and post-EBRT was associated with a significantly higher survival rate regardless of tumor residuals. The restriction of the movement of water molecules depends on several factors related to the aggressiveness of the tumor. When characterizing a tumor with ADC values, the use of a S-ROI, placed on the region showing greatest restriction, probably represents the most aggressive tissue component, analogous to the final histological diagnosis that is based on the most aggressive subregion of the tumor. Our results parallel some previous observations in breast cancer, where ADC measurements with small ROIs were shown to be more accurate than whole-lesion ROIs and were more frequently associated with prognostic factors [30].

Optimization of ADC measurements is of great clinical importance. Bickel et al. [31] concluded that ROI placement significantly affects the ADCs in breast tumors. In many previous studies of cervical cancer, the ADCs were measured on the slice with the largest tumor size using different ROI placement methods, including a single freehand ROI covering the whole tumor, multiple circle-like ROIs of different sizes or three dimensional (3D) -ROIs covering the whole lesion [15, 16]. Importantly, the ADCs differed significantly depending on the measurement procedure, a factor that should be recognized when comparing measurements at different times and in prior studies. In clinical practice, using a L-ROI covering the whole tumor could be less time consuming than using S-ROIs and looking for the region of tumor showing greatest restriction. However, in present study, a greater increase in the mean ADCs depicted using S-ROIs showed strongest association with overall survival. Furthermore, a greater ADC increase between pre- and post-EBRT predicted the recurrence free survival at all timepoints when using S-ROI$_{min}$. The interobserver agreement in our study was good to excellent for both ROI delineation methods, indicating that the S-ROI method can be relied on in clinical work in agreement with the study by Meyer et al. [32]. The equally good reproducibility and the better predictive value thus advocate the use of small ROIs that are placed on the most restricted part of the cervical tumor.

Our study had several limitations. First, the study population was small. Second, due to the retrospective nature of the study, the imaging parameters and MRI scanners varied depending on the hospital where the scans were performed. The strength of our study is that we obtained ADCs at six time points in subgroup of patients, providing new information on the development of ADC values during a treatment protocol combining EBRT and IGBT. Our results suggest that the optimal time to measure and compare an ADC with the pretreatment values is post-ERBT because the change in ADC is already visible and significant and most patients still

have a measurable tumor. Thus, our results could help to schedule the timing of mid-treatment MRI.

## Conclusion

In cervical cancer, ADC values differ significantly depending on the method used to delineate the ROIs, with small ROIs on most restricted areas producing significantly lower ADC values compared to the large whole tumor covering ROIs. Intratumoral ADCs increase significantly between the pretreatment and post-EBRT, as well as between the pretreatment and IGBT MRIs regardless of the ROI delineation method. In clinical practice our results support measuring and comparing the ADC values before treatment and after EBRT, rather than using the ADC values at a single timepoint. Importantly, in our cohort, the greater increase in ADC values between pretreatment and post-EBRT values measured using small standard-size ROIs, seemed to provide a surrogate marker for better overall and recurrence free survival. Standardized methods for timing and delineation of ADC measurements are recommended in future studies.

## Supporting information

**S1 Table. Overview of the imaging acquisition protocols of the diffusion-weighted imaging sequences.**
(DOCX)

**S2 Table. Overall survival, univariate and multivariate analysis.**
(DOCX)

**S3 Table. Recurrence free survival at 3 year timepoint, univariate and multivariate analysis.**
(DOCX)

**S1 Fig. Univariate analysis of cumulative overall survival according to the different prognostic factors for cervical cancer.**
(DOCX)

## Acknowledgments

We thank biostatistician Tuomas Selander for his skillful technical assistance in data analysis.

## Author Contributions

**Conceptualization:** Maarit Anttila, Ritva Vanninen, Auni Lindgren.

**Data curation:** Erikka Holopainen.

**Formal analysis:** Erikka Holopainen, Olli Lahtinen, Auni Lindgren.

**Investigation:** Erikka Holopainen.

**Methodology:** Mervi Könönen, Ritva Vanninen, Auni Lindgren.

**Resources:** Maarit Anttila.

**Software:** Mervi Könönen.

**Supervision:** Ritva Vanninen, Auni Lindgren.

**Writing – original draft:** Erikka Holopainen.

**Writing – review & editing:** Ritva Vanninen, Auni Lindgren.

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
