## [Decision Letter · Decision Letter 0]

16 Jan 2023

PONE-D-22-33558Greater increases in intratumoral Apparent Diffusion Coefficients after chemoradiotherapy predict better overall survival of patients with cervical cancerPLOS ONE

Dear Dr. Holopainen,

Thank you for submitting your manuscript to PLOS ONE. After careful consideration, we feel that it has merit but does not fully meet PLOS ONE’s publication criteria as it currently stands. Therefore, we invite you to submit a revised version of the manuscript that addresses the points raised during the review process.

We look forward to receiving your revised manuscript.

Kind regards,

Kazunori Nagasaka

Academic Editor

PLOS ONE

Journal Requirements:

"This study was supported by the Radiological Society of Finland, Abdominal Radiology Society of Finland, and Kuopio University Hospital Research Foundation."

"E.H: Radiological Society of Finland, no grant number, https://sry.fi

E.H: Abdominal Radiology Society of Finland, no grant number, https://vatsaradiologit.sry.fi

E.H: Kuopio University Hospital Research Foundation, no grant number, https://www.psshp.fi/tutkimussaatio.

The funders had no role in study design, data collection and analysis, decision to publish or preparation of the manuscript."

**Additional Editor Comments:**

Dear Authors,

The manuscript has been peer-reviewed by experts in cervical cancer treatment.

The content is exciting and of clinical significance. We have prepared a rebuttal letter to each comment, and please revise the manuscript accordingly.

Reviewers' comments:

Reviewer's Responses to Questions

**Comments to the Author**

1. Is the manuscript technically sound, and do the data support the conclusions?

Reviewer #1: Yes

Reviewer #2: Yes

2. Has the statistical analysis been performed appropriately and rigorously? 

Reviewer #1: No

Reviewer #2: Yes

3. Have the authors made all data underlying the findings in their manuscript fully available?

Reviewer #1: No

Reviewer #2: Yes

4. Is the manuscript presented in an intelligible fashion and written in standard English?

Reviewer #1: Yes

Reviewer #2: Yes

5. Review Comments to the Author

Reviewer #1: Thank you for reviewing “Greater increases in intratumoral Apparent Diffusion Coefficients after chemoradiotherapy predict better overall survival of patients with cervical cancer”

Major concern:

The focus of your research is excellent. There are no standardized methods for measuring ADCs in patients with cervical cancer, this study indicated the efficacy of monitoring treatment response after chemoradiotherapy if the pre-treatment ADC values are compared to post-treatment values.

However, this is only a univariate analysis, and other confounding factors have not been evaluated, and factors such as tumor size have not been removed. At the very least, a multivariate analysis is necessary, and I urge you to reconsider your analysis method.

Materials and Methods:

{94 patients were excluded because MRI with DWI was not available at all of these time points. Thus, 23 consecutive patients who underwent MRI with DWI at the six consecutive time points were included in the study.}

The lack of MRIs at all 6 time points has resulted in the exclusion of 94 patients and a much lower number of N. Even without MRIs at all 6 time points, is there any way to compare more than 100 patients?

I feel that the drastic decrease in the number of Ns is a bit of a waste in this study.

There seems to be a lot of exclusions due to missing MRI, especially during intra-tissue irradiation. Is there any way to avoid reducing the number of N or subgroup analysis?

Result:

How about adopting FIGO 2018 for the staging classification of stage I B? (ⅠB1-2→1-3)

The statement "ADC remains significantly higher than pre-treatment values during the IGBT period."

However, after IGBT2, both L-ROImean and S-ROImean ADC have decreased and lost significant difference with respect to L-ROI. The S-ROI also barely shows a significant difference. IGBT3 also shows a significant re-elevation in both cases.

In this regard, why does the ADC temporarily decrease after IGBT2?

Can this phenomenon be a limitation in using ADC as a prognostic factor?

Also, you state in the text, "ADC at 3 months post-treatment was not significantly different from that after EBRT."

Why is there no significant difference in ADC when 75% of the patients with residual tumor have disappeared at 3 months after treatment? It seems to me that the shrinkage of the tumor does not correlate with the increase in ADC.

{Smaller tumor size (<4 cm) was associated with lower pretreatment ADCs and the changes in ADCs after EBRT were greater for smaller tumors.}

In considering changes in ADC as a prognostic factor, did you exclude the effect of tumor size? It would make sense if the complete response rate is higher with a higher change in ADC regardless of tumor size.

{An increase in the ADC of >32% between the pre- and post-EBRT scans for L-ROImean predicted better overall survival (P = 0.040). Other ADC values in different time points did not associate with overall or recurrence free survival.}

How can you say that ADC change is a prognostic factor with results that are significant only after EBRT among CCRT and no significant difference in the remaining treatments? This conclusion makes me feel that treatment after EBRT is not effective.

Besides, this is only a univariate analysis, and other confounding factors have not been evaluated, and factors such as tumor size have not been removed. At the very least, a multivariate analysis is necessary, and I urge you to reconsider your analysis method.

Discussion:

{Nakamura et al. [24] reported that the mean ADCs of the primary tumor were significantly associations with FIGO stage, tumor size, stromal and parametrial invasion, lymph node metastasis, lymph vascular space involvement, and adjuvant therapy.}

The opposite results of the present study indicate that the influence of these confounding factors should be eliminated by changing the method of analysis.

{Our results suggest that the predictive information of ADC measurements is greater when the pretreatment and post-EBRT ADCs are compared.}

What is the intent of this sentence? Please provide details.

{In the Kaplan‒Meier univariate analysis of cumulative overall survival, a greater increase (>32%) in the ADC between pre- and post-EBRT was associated with better overall survival (P = 0.040) when using L-ROImean.}

This is only a univariate analysis, and other confounding factors have not been evaluated, and factors such as tumor size have not been removed. At the very least, a multivariate analysis is necessary, and I urge you to reconsider your analysis method.

The evidence showed the other pretreatment, post-ERBT, and post-treatment ADC parameters were not associated with overall or recurrence-free survival.

{Our study had several limitations. First, the study population was small.}

The lack of MRIs at all 6 time points has resulted in the exclusion of 94 patients and a much lower number of N. Even without MRIs at all 6 time points, is there any way to compare more than 100 patients?

Conclusion:

{Importantly, the greater increase in ADCs measured using L-ROIs provided an additional marker for predicting better overall survival.}

The results of this study, statistically, do not suggest this above content as a conclusion.

It is too early to conclude that the greater increase in ADCs is a marker for predicting better overall survival by your analysis method.

Reviewer #2: I am very interested in that the incriesing of ADC after treatment of cervical cancer predicts patient’s prognosis. I would like to know some informations and have a few things to confirm.

Some sentences contain expressions that are little difficult to understand, please fix if possible.

(I can understand what you mean.)

line 26-27 And is this change…

line 251-253 On average,the…, ; There is a description in the previous sentence, but I think it is easier to understand if you describe the measurement tome again.

Line 367-368 Our results suggest that….

line 96, you said “patients with more advanced disease (FIGO stage ⅠB2 or grater) are treated with chemoradiotherapy. But in the clinical scenes, even if the disease is not advanced, surgery may not be performed due to comorbidities, age, and other reasons of the patients.It is better to change the expression because it may cause misunderstanding (Please refer to the cervical cancer treatment guidelines etc.).

line 132,133 ; You wrote “Ninety-four patients”, “23 consecutive patients”

Is there any reason why you changed the numbering method? You should unified numbering method i.e. ;94 patients 23 consecutive patients, Ninety-four patients twenty-three consecutive patients

You wrote grade3 and grade2 in the title of captions Fig.2 and Fig.3. As the histopathological type of cervical cancer, we described as “adenocarcinoma”, “squamous cell carcinoma” without grade, commonly. Please explain specifically.

You measured the ADC of L-ROI from axial images, is not it common to evaluate with sagittal images? If so, is there a reason for that?

In Fig.2, you put on only sagittal images during treatment and after treatment. Isn’t it better to put axial images during and after treatment?

Table 1. Since the stage of cervical cancer has just been revised in FIGO2018, it would be better to state somewhere that the description of the stage follows FIGO2018. If possible, please describe the details about cancer recurrence (recurrent site), and deceased patients.

You wrote that there were no significant changes in the ADCs after IGBT. But in the clinical scenes, IGBT is very effective therapy. Do you have any considerations on no change in ADC after IGBT?

Fig.6

You wrote the ADC changes of stage1~2 was greater than that of progressive cancers. Isn’t it just that the survival rate is better because the red line contains more early stages? I want you to describe the stage of the cancers in the red line and blue line.

6. PLOS authors have the option to publish the peer review history of their article (what does this mean?). If published, this will include your full peer review and any attached files.

Reviewer #1: No

Reviewer #2: **Yes: **Haruka Nishida

---

## [Author Response · Author response to Decision Letter 0]

24 Feb 2023

February 24, 2023

Prof. Kazunori Nagasaka 

Academic Editor

PLOS ONE

Dear Professor Nagasaka,

Please find attached our revised manuscript, PONE-D-22-33558, PLOS ONE, Title: Greater increases in intratumoral Apparent Diffusion Coefficients after chemoradiotherapy predict better overall survival of patients with cervical cancer.

We are grateful for the opportunity to revise our manuscript by taking into account both Reviewer`s insightful and constructive comments and suggestions. We have carefully read all the Editor´s and Reviewer’s comments and modified the manuscript accordingly. We hope that we have addressed all the concerns. The changes are mentioned in our point-by point response as well as highlighted with yellow in the attached revised manuscript labeled as Revised Manuscript with Track Changes. A clean copy of the revised manuscript is also included and labeled as Manuscript.

The funding-related text is removed from the manuscript as requested. The funding statement is now corrected and updated, but unfortunately most of the grands have no grant numbers to be presented.

We hope that you find the responses satisfactory and the manuscript suitable for publication in your distinguished journal. We would also be happy to respond to any further questions and comments that you might have.

Sincerely,

On behalf of all the authors,

Erikka Holopainen

Erikka Holopainen, M.D.

Department of Clinical Radiology

Kuopio University Hospital, Kuopio, Finland

email: erikka.holopainen@pshyvinvointialue.fi

POINT-BY-POINT RESPONSE

Reviewer #1

Major concern:

The focus of your research is excellent. There are no standardized methods for measuring ADCs in patients with cervical cancer, this study indicated the efficacy of monitoring treatment response after chemoradiotherapy if the pre-treatment ADC values are compared to post-treatment values.

However, this is only a univariate analysis, and other confounding factors have not been evaluated, and factors such as tumor size have not been removed. At the very least, a multivariate analysis is necessary, and I urge you to reconsider your analysis method.

Thank you for your kind comment and suggestions to improve the manuscript. We have now performed multivariate analysis and the results are shown in the revised version.

Materials and Methods:

{94 patients were excluded because MRI with DWI was not available at all these time points. Thus, 23 consecutive patients who underwent MRI with DWI at the six consecutive time points were included in the study.}

The lack of MRIs at all 6 time points has resulted in the exclusion of 94 patients and a much lower number of N. Even without MRIs at all 6 time points, is there any way to compare more than 100 patients?

I feel that the drastic decrease in the number of Ns is a bit of a waste in this study.

There seems to be a lot of exclusions due to missing MRI, especially during intra-tissue irradiation. Is there any way to avoid reducing the number of N or subgroup analysis?

Thank you for these important questions. We agree that the number of patients included in the study was low. It was very disappointing to see the lack of MRI scans at all the six timepoints. Our institution is a tertiary care center and patients can come from all over the country to receive IGBT treatments to our center. Not all of them have diffusion weighted images in their pretreatment or post external beam radiotherapy (EBRT) MRI protocols.

Actions taken: Originally, we included consecutive patients with histopathologically proven cervical cancer who were treated at our institution between 2009 and 2020 and required that they had finished their treatment protocol before the end of 2020, to obtain a reasonable follow-up period. In the revised manuscript, we also accepted patients who were diagnosed before the end of 2020 but had not finished their treatment protocol then, as a reasonable follow-up period could now be obtained also for them. Furthermore, we now included all patients who had an MRI scan with diffusion weighted imaging pretreatment and post-EBRT and chemotherapy. Those scans were used in reproducibility analyses and in the univariate and multivariate assessments of prognostic factors. That way we managed to increase the number of patients to 52. Five of those new patients had MRI with DWI at six timepoints and they were included also in the subgroup analysis, increasing the number to 28. Unfortunately, we were not able to reach the 100-patient goal.

Result:

How about adopting FIGO 2018 for the staging classification of stage I B? (ⅠB1-2→1-3)

Thank you for the comment. 

Actions taken: The stage has been changed according to FIGO 2018 classification. See manuscript page 4, line 96 and pages 9-10, Table 1.

The statement "ADC remains significantly higher than pre-treatment values during the IGBT period."

However, after IGBT2, both L-ROImean and S-ROImean ADC have decreased and lost significant difference with respect to L-ROI. The S-ROI also barely shows a significant difference. IGBT3 also shows a significant re-elevation in both cases.

In this regard, why does the ADC temporarily decrease after IGBT2?

Can this phenomenon be a limitation in using ADC as a prognostic factor?

Thank you for the important comment and questions. IGBT is very effective therapy, and often the tumors shrink or even disappear during the treatment. When the tumors get smaller and shrink, they also get denser and this could explain why ADC values decrease at that point, reflecting the restriction of the movement of water molecules. In addition, several of the tumors were so small in the IGBT and post-treatment scans that the ADC measurements could not be performed. In the subgroup analysis, only 14 tumors could be measured at IGBT, compared to 25 tumors measured post-EBRT, possibly leaving the more aggressive ones still visible. The phenomenon that ADC values “fluctuate” during IGBT, can be a limitation in using ADC as a prognostic factor, but this can be avoided if the change in ADC values is measured already after post-EBRT. 

Actions taken: The following sentences have been added to the discussion, page 17, lines 335-340: 

Interestingly, the ADC values were somewhat lower after IGBT2 compared to those post-EBRT. Several of the tumors were so small in the IGBT scans that the ADC measurements could not be performed. In the subgroup analysis, only 14 tumors could be measured at IGBT compared to the 25 tumors measured post-EBRT, possibly leaving the most aggressive ones still visible. In addition, the tumors often shrink during IGBT, and the remaining smaller and denser tumors may in part explain the lower ADC values.

Also, you state in the text, "ADC at 3 months post-treatment was not significantly different from that after EBRT."

Why is there no significant difference in ADC when 75% of the patients with residual tumor have disappeared at 3 months after treatment? It seems to me that the shrinkage of the tumor does not correlate with the increase in ADC.

Thank you for the question. As mentioned above, IGBT is very effective therapy and most of the tumors disappeared during the treatment. In the tumors remaining at 3 moth scans (n = 10) the ADC values showed a significant increase compared to the pretreatment scans which could indicate some cellular response to treatment, while the tumor had somewhat shrunken. There was, however, no difference compared to the post-EBRT scans. Unfortunately, we are not able to fully explain the reason for this phenomenon.

Actions taken: The following sentences have been added to the revised manuscript, page 10, lines 215-219:

In patients with residual tumor after IGBT, the ADCs at 3 moths’ post-treatment MRI were significantly higher than pretreatment ADCs (L-ROImin 843 mm2/s, P = 0.002, S-ROImean 1001 mms2/s, P = 0.008, S-ROImin 884 mm2/s, P = 0.006) but there were no significant changes in the ADCs at 3 months post-treatment compared with post-EBRT.

{Smaller tumor size (<4 cm) was associated with lower pretreatment ADCs and the changes in ADCs after EBRT were greater for smaller tumors.}

In considering changes in ADC as a prognostic factor, did you exclude the effect of tumor size? It would make sense if the complete response rate were higher with a higher change in ADC regardless of tumor size.

Thank you for your valuable comment. We have repeated the univariate calculations now with the larger (n = 52) cohort and some of the results changed. More advanced tumor stage tended to be associated with higher ADCs, larger tumors of over 4 cm had lower ADC values post-EBRT, and tumors that recurred during follow-up had higher ADC values in pretreatment MRI.

Actions taken: Table 3 is now renewed with the larger cohort, pages 13-14. Some new prognostic factors have been added (lymph nodes, parametria invasion and adjuvant therapy).The following sentences are included in the revised manuscript, page 12, lines 257-265: 

Tumor grade or histology were not associated with the pretreatment ADCs. More advanced tumor stage tended to be associated (P = 0.063) with higher ADCs in L-ROImean. Larger tumors of over 4 cm had lower ADC values post EBRT in L-ROImean (P = 0.034) and tended to be lower in S-ROImean (P = 0.076). Tumors that recurred in follow up showed higher ADC values in pretreatment MRI (S-ROImin, P = 0.045; L-ROImin, P = 0.055). In patients, who had adjuvant therapy (n = 5) the increase in ADC values between pre- and post-EBRT did not reach statistical significance using two of the measurement methods (L-ROImean, P = 0.084; S-ROImin, P = 0.105), while in patients with no adjuvant therapy (n = 47) the increase was highly significant with all ROI measurement methods (P = 0.001) (Table 3).

{An increase in the ADC of >32% between the pre- and post-EBRT scans for L-ROImean predicted better overall survival (P = 0.040). Other ADC values in different time points did not associate with overall or recurrence free survival.}

How can you say that ADC change is a prognostic factor with results that are significant only after EBRT among CCRT and no significant difference in the remaining treatments? This conclusion makes me feel that treatment after EBRT is not effective.

Besides, this is only a univariate analysis, and other confounding factors have not been evaluated, and factors such as tumor size have not been removed. At the very least, a multivariate analysis is necessary, and I urge you to reconsider your analysis method.

Thank you for the valuable comments and questions. Our cohort of patients received both concurrent chemoradiotherapy and four cycles of image-guided brachytherapy. After including more patients to the predictive analyses, the increase in ADC values after EBRT remained as a significant prognostic factor. However, this does not mean that the treatment after EBRT is not effective. Rather, this indicates that those patients, whose tumors show an increase in ADC already after EBRT, will benefit most, when they receive also subsequent IGBT. 

Actions taken: We have revised the results with this new larger cohort. The following sentences have been added to page 15, lines 283-842: An increase in the intratumoral ADC > 47 % between pre- and post-EBRT scans for S-ROImin predicted better overall survival (P = 0.018) in Kaplan Meier analyses. 

We calculated the overall and recurrence free survival analysis using also 1-year, 2-year, and 3-year timepoints. To evaluate other confounding factors, we now performed Cox regression multivariate analysis where we included ADC values and their change together with known prognostic factors for cervical cancer. The following sentences have been added to results section, page 15, lines 290-296.

We included ADC values and their change, measured with different ROI delineation methos, together with known prognostic factors of cervical cancer (stage, tumor size, lymph node, parametria invasion and adjuvant therapy) to Cox regression multivariate analysis. For overall survival, at 1-year timepoint pretreatment ADC value remained significant in multivariate analysis (L-ROImean, P = 0.006, HR 61.419 (95% CI 3.19-1181.06). ADC value increase measured with S-ROImin tended to be significant also in multivariate analysis for the whole follow-up time (P = 0.068, HR 0.217 (95% CI 0.04-1.12) and for the 2-year timepoint (P = 0.067, HR 0.138 (95% CI 0.02-1.15).

Discussion:

{Nakamura et al. [24] reported that the mean ADCs of the primary tumor were significantly associations with FIGO stage, tumor size, stromal and parametrial invasion, lymph node metastasis, lymph vascular space involvement, and adjuvant therapy.} 

The opposite results of the present study indicate that the influence of these confounding factors should be eliminated by changing the method of analysis.

Thank you for the comment. We have now reanalyzed our results with known prognostic factors for cervical cancer. Furthermore, the confounding factors are now included in the Cox regression multivariate analysis where we included ADC values and their change, together with known prognostic factors for cervical cancer (see response to the previous point).

Actions taken: Following parameters have been added to Table 3: lymph nodes, parametria invasion and adjuvant therapy. Unfortunately, the lymph vascular space involvement was not available in patient records. The following sentences are now included in the Results section, page 12, lines 257-265: 

Tumor grade or histology were not associated with the pretreatment ADCs. More advanced tumor stage tended to be associated (P = 0.063) with higher ADCs in large ROImean. Larger tumors of over 4 cm had lower ADC values post EBRT in L-ROImean (P = 0.034) and tended to be lower in S-ROImean (P = 0.076). Tumors that recurred in follow up showed higher ADC values in pretreatment MRI (S-ROImin; P = 0.045; L-ROImin, P = 0.055). In patients, who had adjuvant therapy (n = 5) the increase in ADC values between pre- and post-EBRT did not reach statistical significance using two of the measurement methods (L-ROImean, P = 0.084; S-ROImin, P = 0.105), while in patients with no adjuvant therapy (n = 47) the increase was highly significant with all ROI measurement methods (P = 0.001) (Table 3).

The following sentences have been added to the discussion, page 18, lines 349-353: In our cohort, tumor grade, size, histology, parametria invasion or lymph node metastasis were not associated with the pretreatment ADCs while a more advanced tumor stage tended to be associated with ADCs. The somewhat different results are possibly due to the inclusion of different patient populations, variations in ADC measurement techniques, and the small study population.

{Our results suggest that the predictive information of ADC measurements is greater when the pretreatment and post-EBRT ADCs are compared.}

What is the intent of this sentence? Please provide details.

Thank you for the valuable comment. We have repeated the measurements and analysis with a larger number of patients and performed Cox-regression multivariate analysis. The predictive value of ADC difference between pre- and post-EBRT was shown to be statistically significant in 2-year and 3-year timepoints of survival data. A higher pretreatment ADC value predicted better overall survival at 1-year timepoint, but not at 2-year or 3-year timepoints. The post-EBRT ADC-values alone were not significant predictors of survival.

Actions taken: The results are shown in revised manuscript, page 10, lines 207-219 and page 15 lines 283-289 and figure 6.

An increase in the intratumoral ADC of >47% between the pre- and post-EBRT scans for S-ROImin predicted better overall survival (P = 0.018) in Kaplan Meier analyses (Fig. 6). Because of the large variety of follow-up times, we calculated the survival analysis also using 1-year, 2-year, and 3-year timepoints. A higher pretreatment ADC value predicted better overall survival at 1-year timepoint (L-ROImean, P = 0.05). A greater increase of over 47% in the ADC values between pre-and post-EBRT scans predicted better overall survival at 2-year follow-up period (L-ROImean, P = 0.045; S-ROImin, P = 0.008) and at 3-year follow-up period (S-ROI min, P = 0.024).

The following sentence has been added to the discussion page 18, lines 363-365: Our results suggest that the predictive information of ADC measurements is greater when the pretreatment and post-EBRT ADCs are compared, instead of using the ADC values at a single time-point.

{In the Kaplan‒Meier univariate analysis of cumulative overall survival, a greater increase (>32%) in the ADC between pre- and post-EBRT was associated with better overall survival (P = 0.040) when using L-ROImean.}

This is only a univariate analysis, and other confounding factors have not been evaluated, and factors such as tumor size have not been removed. At the very least, a multivariate analysis is necessary, and I urge you to reconsider your analysis method.

The evidence showed the other pretreatment, post-ERBT, and post-treatment ADC parameters were not associated with overall or recurrence-free survival.

Thank you for the valuable comment. We have now performed Cox regression multivariate analysis with the larger cohort. We also calculated the survival using different timepoints (1-year, 2-year and 3 year).

Actions taken: The following sentences have been added in the results section, page 15, lines 290-297:

We included ADC values and their change, measured with different ROI delineation methos, together with known prognostic factors of cervical cancer (stage, tumor size, lymph node, parametria invasion and adjuvant therapy) to Cox regression multivariate analysis. For overall survival, at 1-year timepoint pretreatment ADC value remained significant in multivariate analysis (L-ROImean, P = 0.006, HR 61.419 (95% CI 3.19-1181.06). ADC value increase measured with S-ROImin tended to be significant also in multivariate analysis for the whole follow-up time (P = 0.068, HR 0.217 (95% CI 0.04-1.12) and for the 2-year timepoint (P = 0.067, HR 138 (95% CI 0.02-1.15). None of the other prognostic factors remained statistically significant in multivariate analysis in our cohort.

{Our study had several limitations. First, the study population was small.}

The lack of MRIs at all 6 time points has resulted in the exclusion of 94 patients and a much lower number of N. Even without MRIs at all 6 time points, is there any way to compare more than 100 patients?

Thank you for this important comment. 

Actions taken: We included more patients in the study by including patients diagnosed with cervical cancer by the end of 2020, even if their treatment was undertaken in 2021. We also included now all patients who had pretreatment and post EBRT MRIs with DWI available. We were able to increase the total number of patients to 52. Altogether 28 of these patients had MRIs with DWI available at all six timepoints, so we included them in the subgroup analysis. 

Conclusion:

{Importantly, the greater increase in ADCs measured using L-ROIs provided an additional marker for predicting better overall survival.}

The results of this study, statistically, do not suggest this above content as a conclusion.

It is too early to conclude that the greater increase in ADCs is a marker for predicting better overall survival by your analysis method.

Thank you for the comment. It is true, that our cohort is still small, and the results need confirmation in larger studies. We have also added multivariate analysis now.

Actions taken: We have now modified the conclusion section as follows, page 20, lines 403-411:

In cervical cancer, ADC values differ significantly depending on the method used to delineate the ROIs, with small ROIs on most restricted areas producing significantly lower ADC values compared to the large whole tumor covering ROIs. Intratumoral ADCs increase significantly between the pretreatment, post-EBRT, and IGBT MRIs regardless of the ROI delineation method. In clinical practice our results support measuring and comparing the ADC values before treatment and after EBRT, rather than using the ADC values at a single time-point. Importantly, in our cohort, the greater increase of ADCs after EBRT and measured using small standard-size ROIs seems to provide a surrogate marker for better recurrence free survival and overall survival. Standardized methods for timing and delineation of ADC measurements are recommended in future studies.

In the Abstract, the conclusion has been also modified, page 2, lines 42-44:

ADC values significantly increase during cervical cancer treatment. Greater increases in ADCs after EBRT predicted better overall survival using S-ROIs. Standardized methods for timing and delineation of ADC measurements are advocated in future studies.

Reviewer #2

Some sentences contain expressions that are little difficult to understand, please fix if possible.

(I can understand what you mean.)

line 26-27 And is this change…

Thank you for the comment. We have now tried to rephrase our text.

Actions taken: The following sentences have been added to the abstract, page 2, line 25-27: To evaluate whether 1) the intratumoral apparent diffusion coefficient (ADC) values changes during cervical cancer treatment and 2) are pretreatment ADC values or their change after treatment is predictive for the treatment outcome or overall survival of patients with cervical cancer.

line 251-253 On average,the…, ; There is a description in the previous sentence, but I think it is easier to understand if you describe the measurement tome again.

Thank you for the comment.

Actions taken: The following sentence is added to page 10, lines 210-212: Between pretreatment and post-EBRT MRI, the mean ADCs increased by 26 % for L-ROImean (P < 0.001), by 41 % for S-ROImean (P < 0.001), by 55 % for L-ROImin (P < 0.001) and by 47 % for S-ROImin (P < 0.001).

Line 367-368 Our results suggest that….

Thank you for the observation. We have modified the sentence as follows, page 18, lines 363-365:

Our results suggest that the predictive information of ADC measurements is greater when the pretreatment and post-EBRT ADCs are compared, instead of using the ADC values at a single time-point.

line 96, you said “patients with more advanced disease (FIGO stage ⅠB2 or grater) are treated with chemoradiotherapy. But in the clinical scenes, even if the disease is not advanced, surgery may not be performed due to comorbidities, age, and other reasons of the patients. It is better to change the expression because it may cause misunderstanding (Please refer to the cervical cancer treatment guidelines etc.).

Thank you for your remark. We agree with you on this. The other factors like contraindication to surgery and anesthesia should be considered.

Actions taken: We have reviewed FIGO Cancer Report and corrected the following sentence: page 3, lines 65-69 and reference number10:

 Patients with early-stage disease, a tumor confined to the uterus, or tumors <4 cm in size (FIGO stages IB1, IB2 and IIA1) are usually treated with primary surgical resection and lymphadenectomy, but in cases with contraindications for surgery or anesthesia, chemoradiotherapy is an equally good alternative. Patients with more advanced disease (FIGO stage IB3 or greater) are treated with chemoradiotherapy [4,5,6,10]. 

line 132,133; You wrote “Ninety-four patients”, “23 consecutive patients”

Is there any reason why you changed the numbering method? You should unify numbering method i.e. ;94 patients 23 consecutive patients, Ninety-four patients twenty-three consecutive patients

Thank you for pointing that out. In the beginning of a sentence, we have written the numbers by letters, while inside the sentence by numbers. 

Actions taken: We have now modified the inclusion criteria. the revised text is written as follows, page 4-5, lines 95--105:

Altogether 122 patients staged by pelvic MRI and body CT/PET-CT who were unsuitable for surgical treatment according to the FIGO 2018 staging system received concurrent chemoradiotherapy (CCRT) comprising external-beam radiotherapy (EBRT) (total dose: 45 Gy) with concurrent cisplatin-based chemotherapy and four cycles of image-guided brachytherapy (IGBT) according to the European Society of Gynecological Oncology (ESGO) guidelines [17] and were evaluated as candidates for the present study. Altogether 52 patients (mean age 56 years) had undergone diffusion weighted MRI examinations before treatment and post- EBRT and concurrent chemotherapy and were included in the study. A subgroup of those patients (n = 28, mean age 53 years) underwent consecutive diffusion weighted MRI examinations at six timepoints; 1) pretreatment, 2) post-EBRT and concurrent chemotherapy; 3-5) during IGBT, and 6) the follow-up scan 3 months after the whole treatment (Fig. 1).

You wrote grade3 and grade2 in the title of captions Fig.2 and Fig.3. As the histopathological type of cervical cancer, we described as “adenocarcinoma”, “squamous cell carcinoma” without grade, commonly. Please explain specifically.

Thank you for the comment.

Actions taken: We have removed the grade from captions and included the histopathological type instead, page 6, lines 135 and 144: 

Fig. 2. MRI of a 55-year-old female with stage IIIc1 adenocarcinoma of the cervix.

 Fig. 3. MRI of a 55-year-old female with stage IIIc1 adenocarcinoma of the cervix.

You measured the ADC of L-ROI from axial images, is not it common to evaluate with sagittal images? If so, is there a reason for that?

Thank you for Your question. In our protocol the DWI sequences are obtained only in axial plane and since the slice thickness is 5 mm, sagittal reformats cannot be obtained afterwards. Consequently, the ADC measurements are done in axial plane.

In Fig.2, you put on only sagittal images during treatment and after treatment. Isn’t it better to put axial images during and after treatment?

Thank you for the comment. We feel that sagittal spin echo images were more informative.

Actions taken: We have now included also axial images into the figure, page 6, figure 3.

Table 1. Since the stage of cervical cancer has just been revised in FIGO2018, it would be better to state somewhere that the description of the stage follows FIGO2018. If possible, please describe the details about cancer recurrence (recurrent site), and deceased patients.

Thank you for this relevant comment.

Actions taken: We have included the description FIGO 2018 in the manuscript, page 4, line 96 and Table 1, page 9. 

We have also included available details about cancer recurrence sites. Since our institution is a tertiary care center, and patients can come from all over the country for IGBT treatment, we don’t have information about the deceased patient other than the date of death. 

Action taken: The following sentences are added to page 8, lines 178-181. Twenty patients had cancer recurrence and 12 had progressive disease. The most common site for recurrence was lymph nodes following lungs and brain. During the follow up period from the time of diagnosis until end of year 2022, 17 patients had died.

You wrote that there were no significant changes in the ADCs after IGBT. But in the clinical scenes, IGBT is very effective therapy. Do you have any considerations on no change in ADC after IGBT?

Thank you for the important comment. As you stated, IGBT is very effective therapy, and in our cohort most of the tumors tend to disappear during the treatment cycle. Several of the tumors were so small in the IGBT scans that the ADC measurements could not be performed. During the IGBT, ADC values seemed to decrease little in the remaining tumors. We think this happened because the remaining tumors were the more aggressive ones and second, the tumor shrinks. As the treatment continues, ADC values remain relatively the same. If the ADC values during IGBT are compared to ADC values of a previous scan, there is no statistically significant difference but if the ADC values are compared to pre-treatment ADC values, the increase remains statistically significant.

Actions taken: The following sentences have been added to the discussion, page 17, lines 335-340:

Interestingly, the ADC values were somewhat lower after at IGBT2 compared to those post-EBRT. Several of the tumors were so small in the IGBT scans that the ADC measurements could not be performed. In the subgroup analysis, only 14 tumors could be measured at IGBT compared to the 25 tumors measured post-EBRT, possibly leaving the most aggressive ones still visible. In addition, the tumors often shrink during IGBT, and the remaining smaller and denser tumors may in part explain the lower ADC values.

Fig.6

You wrote the ADC changes of stage1~2 was greater than that of progressive cancers. Isn’t it just that the survival rate is better because the red line contains more early stages? I want you to describe the stage of the cancers in the red line and blue line.

Thank you for the comment. Since we included more patients to the study, we calculated the survival with increased number of patients and with multivariate analysis. We included ADC values and their change, measured with different ROI delineation methods, together with known prognostic factors of cervical cancer (stage, tumor size, lymph node, parametria invasion and adjuvant therapy) to Cox regression multivariate analysis, but none of the prognostic factors remained statistically significant, ADC value for S-ROImin tended to be significant (P = 0.068, HR 0.217 (95% CI 0.04-1.12).

Actions taken: We updated the changes in the manuscript accordingly: pages 15-16, lines 290–305 and Fig. 6. In the Figure, the univariate analysis between two groups indicates patients with different levels of ADC increase. The Figure legend is now as follows, page 16, lines 307-311: 

Fig. 6. Univariate analysis of cumulative overall survival according to the relative change in ADC values from pretreatment to post-EBRT MRI scan using intratumoral S-ROIs.

Blue line indicates group of patients with an increase of over 47% in ADC values (n = 14). Red line indicates group of patients with an increase of less than 47% in ADC values (n = 23) An increase in the ADC of over 47% for S-ROI predicted better overall survival (P = 0.018).

---

## [Decision Letter · Decision Letter 1]

12 Mar 2023

PONE-D-22-33558R1Greater increases in intratumoral Apparent Diffusion Coefficients after chemoradiotherapy predict better overall survival of patients with cervical cancerPLOS ONE

Dear Dr. Holopainen,

Thank you for submitting your manuscript to PLOS ONE. After careful consideration, we feel that it has merit but does not fully meet PLOS ONE’s publication criteria as it currently stands. Therefore, we invite you to submit a revised version of the manuscript that addresses the points raised during the review process. Please submit your revised manuscript by Apr 26 2023 11:59PM. If you will need more time than this to complete your revisions, please reply to this message or contact the journal office at plosone@plos.org. Please include the following items when submitting your revised manuscript:A rebuttal letter that responds to each point raised by the academic editor and reviewer(s). You should upload this letter as a separate file labeled 'Response to Reviewers'.A marked-up copy of your manuscript that highlights changes made to the original version. You should upload this as a separate file labeled 'Revised Manuscript with Track Changes'.An unmarked version of your revised paper without tracked changes. You should upload this as a separate file labeled 'Manuscript'.If applicable, we recommend that you deposit your laboratory protocols in protocols.io to enhance the reproducibility of your results. Protocols.io assigns your protocol its own identifier (DOI) so that it can be cited independently in the future. For instructions see: https://journals.plos.org/plosone/s/submission-guidelines#loc-laboratory-protocols. Additionally, PLOS ONE offers an option for publishing peer-reviewed Lab Protocol articles, which describe protocols hosted on protocols.io. Read more information on sharing protocols at https://plos.org/protocols?utm_medium=editorial-email&utm_source=authorletters&utm_campaign=protocols.

We look forward to receiving your revised manuscript.

Kind regards,

Kazunori Nagasaka

Academic Editor

PLOS ONE

Journal Requirements:

Additional Editor Comments:

Dear Authors,

Please add some information regarding the multivariate analysis, and then, we will accept your manuscript for publication.

Reviewers' comments:

Reviewer's Responses to Questions

**Comments to the Author**

1. If the authors have adequately addressed your comments raised in a previous round of review and you feel that this manuscript is now acceptable for publication, you may indicate that here to bypass the “Comments to the Author” section, enter your conflict of interest statement in the “Confidential to Editor” section, and submit your "Accept" recommendation.

Reviewer #1: All comments have been addressed

Reviewer #2: All comments have been addressed

2. Is the manuscript technically sound, and do the data support the conclusions?

Reviewer #1: Yes

Reviewer #2: Yes

3. Has the statistical analysis been performed appropriately and rigorously? 

Reviewer #1: Yes

Reviewer #2: Yes

4. Have the authors made all data underlying the findings in their manuscript fully available?

Reviewer #1: Yes

Reviewer #2: Yes

5. Is the manuscript presented in an intelligible fashion and written in standard English?

Reviewer #1: Yes

Reviewer #2: Yes

6. Review Comments to the Author

Reviewer #1: Major point

Multivariate analysis has been added and good results have been obtained, however there are no Figures and Tables for multivariate analysis, please add them. It is difficult to convey this information in text only.

Since the results of the multivariate analysis are the most important in this study, I think it would be better to include all the results of the multivariate analysis in a Figure and Table, even if it is supplemental.

Please include not only the prognostic factors that showed significant differences, but also those that did not show significant differences.

Please include not only p-value and Hazard Ratio but also the actual values for the entire analysis. The curves from the univariate analysis alone are not convincing.

Minor point

{Using Cox regression multivariate analysis ADC value change measured with LROImin was statistically significant at 3-year timepoint (P = 0.033, HR 0.204 (95% CI 0.05-0.88) and pretreatment ADC value S-ROImin (P = 0.05, HR 0.340 (95%CI 0.12-1.00) at 3-year timepoint remained also significant.}

→In principle, do you recommend that both S-ROI and L-ROI be measured, with L-ROI as the percentage of ADC value change before and after treatment, and S-ROI as the pretreatment ADC value as a prognostic predictor at 3-year timepoint?

Please describe the recommended use as a prognostic factor.

In the Univariate analysis curve (Fig. 6), the number of N is 37 (14+23), why did 15 cases drop out of 52? Please describe the reason.

After EBRT, ADC could be measured in only 14 of 28 cases due to tumor shrinkage.

I question whether the use of ADC value change immediately after EBRT as a prognostic factor after CCRT, even though the number of cases after EBRT has decreased by half.

Why not analyze only the 14 cases that could be measured ADC all 6 times?

Only 10 patients had residual tumor at 3 months, and in those 10 patients, there was no significant difference in ADC value from EBRT to IGBT.

Is the prognosis better for patients with a larger ADC change value in both residual tumor group and no residual tumor group?

Can you conclude that a larger ADC change rate between pre-treatment and post-EBRT is associated with a significantly higher survival rate regardless of tumor residuals?

In this multivariate analysis, you state that low pre-treatment ADC and high ADC change value were the sole prognostic factors, please add that data not only in the text, but also in Figure, Table. Please also add to in Figure, Table the known prognostic factors for cervical cancer (stage, tumor size, lymph nodes, parametrial invasion, and postoperative adjuvant therapy) which were no significant differences. The results of this multivariate analysis are most important for this study.

Discussion:

Of the 28 cases included in the subgroup analysis, the number was reduced to 25 after EBRT. Why did 3 cases drop out? I don't understand why 3 patients dropped out since MRI could have been performed at all 6 points. Please explain.

I understand that 14 cases could not be measured for ADC after EBRT because of small residual tumor, but I do not understand why 3 cases dropped out, assuming that the remaining 11 cases could be measured for ADC.

The 14 patients in the group who had ADC measured at all 6 times (residual tumor group) and the 14 patients in the group who had no ADC measured after IGBT (no residual tumor group) should be compared, respectively.

If the rate of increase in ADC after EBRT is significant in both groups, then the rate of increase after EBRT may be able to predict prognosis to some extent, regardless of residual tumor.

{Thus, our results could help to schedule the timing of mid-treatment MRI.}

→What does this mean?

Based on the results of this study, I believe that MRI should be performed before the residual tumor disappears for prognostic purposes, and based on the results of multivariate analysis, it should be performed immediately after EBRT.

I think that prognosis cannot be predicted without calculating the ADC change value between pre-treatment and post-EBRT, which was significantly different.

{The predictive value of the ADC difference before and after EBRT was shown to be statistically significant at the 2 and 3 year timepoints of the survival data} stated.

→Is the cutoff value for this significant difference also 47%?

{Higher pre-treatment ADC values predicted better overall survival at 1 year, but not at 2 or 3 years} stated.

→What is the cutoff value of the pre-treatment ADC value when a significant difference is found? Also, if the cutoff value is lowered, is there likely to be a significant difference at timepoints 2 and 3 years?

Conclusion:

{Intratumoral ADCs increase significantly between the pretreatment, post-EBRT, and IGBT MRIs regardless of the ROI delineation method.}

→Only ”pre-treatment and post-EBRT” and “pre-treatment and post-IGBT” were significant differences. Please clarify the description.

Please provide the recommended use of ADC measurement as a prognostic factor based on the results of this study.

Reviewer #2: (No Response)

7. PLOS authors have the option to publish the peer review history of their article (what does this mean?). If published, this will include your full peer review and any attached files.

Reviewer #1: No

Reviewer #2: No

---

## [Author Response · Author response to Decision Letter 1]

12 Apr 2023

POINT BY POINT RESPONSE TO THE REVIEWER

Reviewer #1: Major point

Multivariate analysis has been added and good results have been obtained, however there are no Figures and Tables for multivariate analysis, please add them. It is difficult to convey this information in text only.

Since the results of the multivariate analysis are the most important in this study, I think it would be better to include all the results of the multivariate analysis in a Figure and Table, even if it is supplemental.

Please include not only the prognostic factors that showed significant differences, but also those that did not show significant differences.

Please include not only p-value and Hazard Ratio but also the actual values for the entire analysis. The curves from the univariate analysis alone are not convincing.

Thank you for all the valuable comments and suggestions to improve the manuscript. We have added Supplementary tables and figures of the multivariate analyses for overall survival and recurrence free survival. The changes are shown in the revised version of our manuscript.

Action taken: See supplementary tables S2, S3 and supplementary figure S4.

Minor point

{Using Cox regression multivariate analysis ADC value change measured with LROImin was statistically significant at 3-year timepoint (P = 0.033, HR 0.204 (95% CI 0.05-0.88) and pretreatment ADC value S-ROImin (P = 0.05, HR 0.340 (95%CI 0.12-1.00) at 3-year timepoint remained also significant.}

→In principle, do you recommend that both S-ROI and L-ROI be measured, with L-ROI as the percentage of ADC value change before and after treatment, and S-ROI as the pretreatment ADC value as a prognostic predictor at 3-year timepoint?

Please describe the recommended use as a prognostic factor.

Thank you for the question. We used different methods in our measurements and assessed their prognostic value in several univariate and multivariate analyses at different timepoints. Although the L-ROI measurements provided significant or almost significant results in some analyses, the S-ROI measurements provided significant results in a larger number of analyses. ∆ ADC between pretreatment and post-EBRT ADC measurements proved to be a significant prognostic predictor in OS and RFS calculations. Based on those results we feel that the best method is to use the small ROI minimum value since that gave us best overall results. 

For detailed results, we refer to the sentences in the manuscript, see page 15-16, lines 283-307.

Actions taken: The following sentence has been added to conclusion section of the manuscript, see page 20, lines 420-422: 

Importantly, in our cohort, the greater increase in ADC values between pretreatment and post-EBRT values measured using small standard-size ROIs, seemed to provide a surrogate marker for better overall and recurrence free survival.

In the Univariate analysis curve (Fig. 6), the number of N is 37 (14+23), why did 15 cases drop out of 52? Please describe the reason.

Thank you for the comment. The 15 cases out of the whole group of 52 dropped out since the tumors had vanished already during the EBRT and there was no visible tumor in the post-EBRT scans for ADC measurement.

Actions taken: The following sentence in the manuscript has been modified, see page 8, lines 183-185: Following EBRT and concurrent cisplatin chemotherapy, 37 (71%) of the 52 patients had a measurable residual tumor on the post-EBRT MRI and 15 (29%) of the tumors had vanished during the chemoradiotherapy.

After EBRT, ADC could be measured in only 14 of 28 cases due to tumor shrinkage.

I question whether the use of ADC value change immediately after EBRT as a prognostic factor after CCRT, even though the number of cases after EBRT has decreased by half.

Why not analyze only the 14 cases that could be measured ADC all 6 times?

Only 10 patients had residual tumor at 3 months, and in those 10 patients, there was no significant difference in ADC value from EBRT to IGBT.

Is the prognosis better for patients with a larger ADC change value in both residual tumor group and no residual tumor group?

Can you conclude that a larger ADC change rate between pre-treatment and post-EBRT is associated with a significantly higher survival rate regardless of tumor residuals?

Thank you for the comment. We are sorry, that we had not been clear in our description of the results. We have measured the ADC values from 37 patients after EBRT. During chemoradiotherapy 15 tumors of the original 52 tumors had shrinked or vanished, and ADC was therefore unmeasurable after chemoradiotherapy. During the intracavitary treatment the healing continued and in the end of the whole treatment (at 3month scan) only 10 tumors out of 52 were still visible and measurable.

Action taken: The following sentences have been added to the chapter ‘ADCs and their changes during treatment’ see page10, lines 215-222: 

To further analyze the importance of a residual tumor after treatments, the patients with a measurable tumor in the post-EBRT scans (n =37) were divided to those with a residual tumor in the 3-month post-treatment scan (n = 10) and those with no residual tumor in the 3-month post-treatment scan (n = 27). The ADCs were significantly higher at the post-EBRT measurements than in the pretreatment scans in both groups (L-ROImean P = 0.006, S-ROImin P = 0.017 in residual tumor group and L-ROImean P < 0.001, S-ROImin P < 0.001 in no residual tumor group respectively. The ADC change was also significant for L-ROImin and S-ROImean in both groups). 

(The detailed results were the following: In residual tumor group the mean pretreatment intratumoral ADC was 883 mm2/s for L-ROImean and 580 mm2/s for S-ROImin. The post-EBRT ADC values were 1097 mm2/s (P = 0.006) for L-ROImean and 751 mm2/s (P = 0.017) for S-ROImin. In no residual tumor group the ADC values were before and after EBRT 875 and 1120 mm2/s (P < 0.001) for L-ROImean and 600 and 879 mm2/s (P < 0.001) for S-ROImin respectively. The ADC change was also significant for L-ROImin and S-ROImean in both groups.)

To further analyze the prognostic importance of the ADC change in the presence or absence of a post treatment residual tumor, we now made a new variable ‘’Remaining residual tumor after treatments’’ and included this for Cox regression multivariate analyses for overall survival calculations. Larger ADC change rate between pre-treatment and post-EBRT remained still statistically significant, regardless of a tumor residual. 

Action taken: After including the “Residual tumor” parameter in the Cox regression multivariate analyses for Overall survival, we have checked the results in the manuscript and included the parameters in the supplementary tables. See page 15, lines 292-298 and Supplementary table S2.

The following sentence has been added to the discussion, see page 19, lines 378-379: A larger ADC change rate between pre-treatment and post-EBRT was associated with a significantly higher survival rate regardless of tumor residuals.

In this multivariate analysis, you state that low pre-treatment ADC and high ADC change value were the sole prognostic factors, please add that data not only in the text, but also in Figure, Table. Please also add to in Figure, Table the known prognostic factors for cervical cancer (stage, tumor size, lymph nodes, parametrial invasion, and postoperative adjuvant therapy) which were no significant differences. The results of this multivariate analysis are most important for this study.

Thank you for the comments. Our Cox regression multivariate analysis results are shown in supplementary tables 2 and 3. S2 table shows overall survival with univariate and multivariate analysis with prognostic factors for cervical cancer (stage, tumor size, lymph nodes, parametria invasion and adjuvant therapy). As described above, we also used the residual tumor variable, but it did not bring any advantage. S3 table shows recurrence free survival at 3 year timepoint. In both univariate and multivariate analysis, the ADC change provided significant prognostic value at that timepoint, but none of the clinical variables remained significant in this analysis.

In addition, the supplementary figure S4 illustrates that the known prognostic clinical factors for cervical cancer did not have statistically significant prognostic value in our cohort. This may be due to the small patient population of our study. We submit this new Figure as a supplementary figure. Figure 6 reflects the overall survival and remains as a separate Figure in the manuscript.

Supplementary Figure. Univariate analysis of cumulative overall survival according to the different prognostic factors for cervical cancer.

A: FIGO stage I-IV (P = 0.830): blue line indicates stage I, red stage II, green stage III, and orange stage IV.

B: Tumor size (P = 0.067): blue line indicates tumor size < 4 cm and red line ≥ 4 cm.

C: Lymph nodes (P = 0.239): blue line indicates negative lymph nodes and red line positive lymph nodes.

D: Parametria invasion (P = 0.177): blue line indicates parametria invasion, red line no parametria invasion.

E: Adjuvant therapy (P = 0.295): blue line indicates that patient received adjuvant therapy, red line indicates no adjuvant therapy.

F: Residual tumor (P = 0.257): blue line indicates no residual tumor at 3 months post-treatment MRI and red line indicates residual tumor at 3 month post-treatment MRI

Discussion:

Of the 28 cases included in the subgroup analysis, the number was reduced to 25 after EBRT. Why did 3 cases drop out? I don't understand why 3 patients dropped out since MRI could have been performed at all 6 points. Please explain.

I understand that 14 cases could not be measured for ADC after EBRT because of small residual tumor, but I do not understand why 3 cases dropped out, assuming that the remaining 11 cases could be measured for ADC.

Thank you for the question. We are sorry, that we have not been clear in our description of the results. In the subgroup of 28 patients who had MRIs that included DWI and ADC at all six timepoints, 3 of them had no visible tumor after EBRT although DWI and ADC sequences were taken. That is the reason why they dropped out. 25 out of 28 were measured post-EBRT. And later, during brachy therapy when the more excellent effect of the treatment was achieved, more cases dropped out and there was no visible tumor to be measured. Finally, in the 3 month post-treatment imaging only 10 cases had a measurable residual tumor.

The 14 patients in the group who had ADC measured at all 6 times (residual tumor group) and the 14 patients in the group who had no ADC measured after IGBT (no residual tumor group) should be compared, respectively.

If the rate of increase in ADC after EBRT is significant in both groups, then the rate of increase after EBRT may be able to predict prognosis to some extent, regardless of residual tumor.

Thank you for this insightful comment. See above our response to the previous comment by the Reviewer. We have the EBRT results from 37patients out of 52. So, we used the whole cohort not only the subgroup of patients. We made a new variable by dividing our cohort into residual tumor groups where there was a residual tumor seen at 3 months post-treatment scan (n = 10) and no residual tumor group (n = 27). We analyzed the ADC change between pre-treatment and post-EBRT scans. There was a significant difference between pre- and post-EBRT ADC values for both groups, regardless of residual tumor. 

Actions taken: The following sentences are added to the manuscript, page 10, lines 215-222: To further analyze the importance of a residual tumor after treatments, the patients with a measurable tumor in the post-EBRT scans (n =37) were divided to those with a residual tumor in the 3-month post-treatment scan (n = 10) and those with no residual tumor in the 3-month post-treatment scan (n = 27). The ADCs were significantly higher at the post-EBRT measurements than in the pretreatment scans in both groups (L-ROImean P = 0.006, S-ROImin P = 0.017 in residual tumor group and L-ROImean P < 0.001, S-ROImin P < 0.001 in no residual tumor group respectively. The ADC change was also significant for L-ROImin and S-ROImean in both groups).

We also included this new variable “residual tumor after treatment” in the Cox regression multivariate analysis but the results remained almost same. See supplementary table S2.

The following sentence has been added to the discussion, see page 19, line 378-379: A larger ADC change rate between pre-treatment and post-EBRT was associated with a significantly higher survival rate regardless of tumor residuals.

{Thus, our results could help to schedule the timing of mid-treatment MRI.}

→What does this mean?

Based on the results of this study, I believe that MRI should be performed before the residual tumor disappears for prognostic purposes, and based on the results of multivariate analysis, it should be performed immediately after EBRT.

I think that prognosis cannot be predicted without calculating the ADC change value between pre-treatment and post-EBRT, which was significantly different.

Thank you for the important comment. That was exactly what we wanted to say. The best time for mid-treatment ADC value measurement is post-EBRT because then most of the tumors are still visible and measurable and the change in ADC is most obvious. During brachy therapy the tumors shrink and even disappear, and there might not be anything to measure and if there is a visible tumor, ADC values do not change so dramatically anymore.

Actions taken: The following sentences have been added to Discussion, see page 18, lines 349-352: Based on the results of this study, mid-treatment MRI should be performed before the residual tumor disappears for prognostic purposes, and based on the results of multivariate analysis, it should be performed immediately after EBRT. Prognosis can be predicted by calculating the ADC change value between pre-treatment and post-EBRT.

{The predictive value of the ADC difference before and after EBRT was shown to be statistically significant at the 2 and 3 year timepoints of the survival data} stated.

→Is the cutoff value for this significant difference also 47%?

Thank you for the important question. The 47 % is the average difference measured in percent between mean pretreatment and mean Post-EBRT ADCs for S-ROImin. The cut-off value for S-ROImin change between pretreatment and post-EBRT ADC that we got from ROC curve using Youden index was 197,5 mm2/s. We got the same cut-off value also for 2- and 3-year time points. For L-ROImean (difference ADC) the cut-off value was 247,5 mm2/s.

Actions taken: The following sentences have been modified, see page 16, lines 312-317: ADC value difference between the pretreatment and post-EBRT scans was dichotomized using ROC curve to define the Youden index providing the cutoff value of 197.5 mm2/s. Blue line indicates group of patients with an increase of over 197.5 mm2/s in ADC values (n = 14). Red line indicates group of patients with an increase of less than 197.5 mm2/s in ADC values (n = 23) An increase in the ADC of over 197.5 mm2/s for S-ROImin predicted better overall survival (P = 0.018).

{Higher pre-treatment ADC values predicted better overall survival at 1 year, but not at 2 or 3 years} stated.

→What is the cutoff value of the pre-treatment ADC value when a significant difference is found? Also, if the cutoff value is lowered, is there likely to be a significant difference at timepoints 2 and 3 years?

Thank you for the comment. ROC-curve analysis was used to determine the Youden-index, which was used as a cut-off value for dichotomization for each of the timepoints separately. Only pretreatment ADC predicted better overall survival at 1-year timepoint (L-ROImean cutoff = 791,0 mm2/s)(P = 0.05). We repeated the calculations using the 1-year timepoint cutoff (791,0 mm2/s) for 2- and 3-year survival analysis, but there were no significant results (2-year timepoint P = 0.113, and 3-year timepoint P = 0.165).

Conclusion:

{Intratumoral ADCs increase significantly between the pretreatment, post-EBRT, and IGBT MRIs regardless of the ROI delineation method.}

→Only ”pre-treatment and post-EBRT” and “pre-treatment and post-IGBT” were significant differences. Please clarify the description.

Please provide the recommended use of ADC measurement as a prognostic factor based on the results of this study.

Thank you for the important comment. We clarify our conclusion and want to underly that the use of small ROI is more convenient because it may serve as a prognostic marker for survival.

Actions taken: We have added the following sentences to Discussion, page 18, lines 349-352: Based on the results of this study, mid-treatment MRI should be performed before the residual tumor disappears for prognostic purposes, and based on the results of multivariate analysis, it should be performed immediately after EBRT. Prognosis can be predicted by calculating the ADC change value between pre-treatment and post-EBRT.

In addition, we have now modified the following sentences: See page 20, lines 416-418: Intratumoral ADCs increase significantly between the pretreatment and post-EBRT as well as between the pretreatment and IGBT MRIs regardless of the ROI delineation method.

And page 20, lines 420-422, Conclusion: Importantly, in our cohort, the greater increase in ADC values between pretreatment and post-EBRT values measured using small standard-size ROIs, seemed to provide a surrogate marker for better overall and recurrence free survival.

---

## [Editor Report · Decision Letter 2]

2 May 2023

Greater increases in intratumoral Apparent Diffusion Coefficients after chemoradiotherapy predict better overall survival of patients with cervical cancer

PONE-D-22-33558R2

Dear Dr. Holopainen,

We’re pleased to inform you that your manuscript has been judged scientifically suitable for publication and will be formally accepted for publication once it meets all outstanding technical requirements.

Kind regards,

Kazunori Nagasaka

Academic Editor

PLOS ONE

Additional Editor Comments (optional):

Dear Authors,

Thank you very much for your submission to Plos One.

Indeed, it is very intriguing to analyze the intratumor ADCs before treatment and after EBRT.

After careful revision, we consider the manuscript worth publishing in our journal.

Again, thank you very much for your submission.

Sincerely,

Kazunori Nagasaka
---

## [Editor Report · Acceptance letter]

4 May 2023

PONE-D-22-33558R2 

Greater increases in intratumoral Apparent Diffusion Coefficients after chemoradiotherapy predict better overall survival of patients with cervical cancer 

Dear Dr. Holopainen:

I'm pleased to inform you that your manuscript has been deemed suitable for publication in PLOS ONE. Congratulations! Your manuscript is now with our production department. 

Kind regards, 

on behalf of

Professor Kazunori Nagasaka 

Academic Editor

PLOS ONE